# Continual Gaussian Mixture Distribution Modeling for Class Incremental Semantic Segmentation

**Guilin Zhu**[1]  **Runmin Wang**[2]  **Yuanjie Shao**[1]  **Weidong Yang**[1,*]

**Nong Sang**[1]  **Changxin Gao**[1,*]

[1]National Key Laboratory of Multispectral Information Intelligent Processing Technology,
School of Artificial Intelligence and Automation, Huazhong University of Science and Technology,
[2]School of Information Science and Engineering, Hunan Normal University
{gzhu, shaoyuanjie,Yangwd,nsang,cgao}@hust.edu.cn,
runminwang@hunnu.edu.cn

## Abstract

Class incremental semantic segmentation (CISS) enables a model to continually segment new classes from non-stationary data while preserving previously learned knowledge. Recent top-performing approaches are prototype-based methods that assign a prototype to each learned class to reproduce previous knowledge. However, modeling each class distribution relying on only a single prototype, which remains fixed throughout the incremental process, presents two key limitations: (i) a single prototype is insufficient to accurately represent the complete class distribution when incoming data stream for a class is naturally multimodal; (ii) the features of old classes may exhibit anisotropy during the incremental process, preventing fixed prototypes from faithfully reproducing the matched distribution. To address the aforementioned limitations, we propose a Continual Gaussian Mixture Distribution (CoGaMiD) modeling method. Specifically, the means and covariance matrices of the Gaussian Mixture Models (GMMs) are estimated to model the complete feature distributions of learned classes. These GMMs are stored to generate pseudo-features that support the learning of novel classes in incremental steps. Moreover, we introduce a Dynamic Adjustment (DA) strategy that utilizes the features of previous classes within incoming data streams to update the stored GMMs. This adaptive update mitigates the mismatch between fixed GMMs and continually evolving distributions. Furthermore, a Gaussian-based Representation Constraint (GRC) loss is proposed to enhance the discriminability of new classes, avoiding confusion between new and old classes. Extensive experiments on Pascal VOC and ADE20K show that our method achieves superior performance compared to previous methods, especially in more challenging long-term incremental scenarios. The source code is available at https://github.com/zhu-gl-ux/CoGaMiD

## 1 Introduction

Semantic segmentation has witnessed remarkable advancements over the decades, particularly with the emergence of deep learning. A wide range of architectures, such as convolution neural networks (CNNs) [23, 18] and Transformers [12, 42, 26], have been developed to tackle the challenge of semantic segmentation. Traditional semantic segmentation networks acquire the ability to handle a predefined number of semantic categories by conducting a one-time training process. However, in realistic applications, the trained segmentation model is expected to learn new concepts continuously under the situation that the previously labeled data are not available due to privacy or legal reasons.

---

[*]Co-corresponding Authors.

39th Conference on Neural Information Processing Systems (NeurIPS 2025).

Fine-tuning the old model with new data results in overfitting the new classes quickly while forgetting learned old classes, termed catastrophic forgetting [29].

To alleviate this problem, equipping segmentation models with the ability to continually learn new classes from the non-stationary data while preserving previously learned classes has emerged as a pivotal task, known as class incremental semantic segmentation (CISS). Classically, studies in CISS explore various techniques, such as knowledge distillation [30, 44, 1, 34], pseudo-labeling [13, 50, 4], contrastive learning [48, 55, 49], and proposal generation [5, 52, 53] to maintain a balance between learning new knowledge and preserving old knowledge. Although impressive, these methods [34, 13, 55, 52] still suffer from catastrophic forgetting, leading to a performance degradation in the old classes, especially in the challenging long-term incremental scenarios.

Recent advances in CISS [60, 11, 7] have adopted a prototype-based technique to incorporate knowledge from previous classes into the new model, achieving a good balance between stability and plasticity. In comparison to previous exemplar-replay methods [28, 34, 5], these methods effectively mitigate privacy concerns and storage costs by maintaining a single prototype for each learned class. Although prototype-based replay methods have demonstrated remarkable performance, certain limitations still persist. Firstly, accurately estimating the distribution of each learned class is essential for preserving previously acquired knowledge. However, a single prototype, represented by the mean of the features and the corresponding variances, is insufficient to precisely capture the complete class distribution when the data streams for a class is inherently multimodal. Secondly, the distributions of learned classes may exhibit anisotropy, since the features appear shift in various directions due to the continual updates of model parameters during the incremental process [16]. Nevertheless, the class prototypes are stored in an unaltered manner, which prevents the faithful reproduction of the corresponding distributions in the feature space.

To this end, we propose a novel method, called Continual Gaussian Mixture Distribution (CoGaMiD) modeling, to address the limitations of current features replay solutions from a prototype perspective. Specifically, after each training step, the feature extractor is frozen to extract class features from current training data. The distribution of each learned class is estimated using an independent Gaussian mixture model with multiple Gaussian components. Utilizing these GMMs, CoGaMiD generates more accurate pseudo-features that represent the distributions of old classes to support the new model in distinguishing between novel and old classes. During the incremental learning process, the features of old classes may exhibit anisotropy due to the model training, a portion of old class distributions will deviate from the saved GMMs, leading to the issue of mismatch. To address this, a Dynamic Adjustment (DA) strategy is developed to ensure that the generated pseudo-features can continuously adapt to the distribution anisotropy by updating the stored GMMs with features of previous classes in current data steams. Additionally, maintaining discriminative representations is essential for accurately learning and estimating the distribution of novel classes. Therefore, we introduce a Gaussian-based Representation Constraint (GRC) loss to push the features of new classes from the centroids of the similar old ones. The contributions of this work are summarized as:

- We present the Continual Gaussian Mixture Distribution (CoGaMiD) modeling, a method that incrementally learns novel classes by generating pseudo-features of learned classes using GMMs. Compared to single prototype-based CISS methods, our approach more effectively reproduces the precise distribution of old knowledge by leveraging the inherent advantages of GMMs in modeling distributions from a multimodal perspective.

- We designed a novel Dynamic Adjustment (DA) strategy to enable the generated pseudo-features continuously adapt to the anisotropic features during the incremental process. Moreover, a Gaussian-based Representation Constraint (GRC) loss is proposed to maintain a discriminative distance between new and old classes.

- Through extensive experiments on PASCAL VOC [15] and ADE20K [57], we demonstrate the state-of-the-art performance of our method across various CISS scenarios, particularly in more challenging long-term incremental scenarios.

## 2 Related Work

### 2.1 Class Incremental Learning

Class Incremental Learning (CIL) aims to overcome the limitations of traditional model training by continuously acquiring new concepts while retaining previously learned knowledge. The main challenge that CIL systems will encounter here is catastrophic forgetting [29]. Existing methods primarily address this problem through three main approaches: regularization, replay, and parameter isolation. In regularization-based methods, additional regularization terms [21, 17, 39, 36] or knowledge distillation [45, 59, 14] are introduced to prevent significant changes to the knowledge of previously learned classes in the new model. Replay-based methods store [35, 27, 2] or generate [40, 19] a small sample set of previous tasks and replay them when learning the new tasks. Parameter isolation methods [43, 25, 20] assign dedicated parameter modules to new tasks, while keeping the parameters of old tasks fixed to be protected from forgetting.

### 2.2 Gaussian Mixture Models in CIL

A series of studies have investigated the use of Gaussian Mixture Models (GMMs) in the CIL task. [58] pre-assigns multiple virtual embeddings using GMMs to prepare the model for future classes. [54] introduces GMMs to automatically and simultaneously estimate both prototype number and prototypes. [33, 41] employ GMMs to generate class-specific images. However, their applications have been largely restricted to simpler datasets, such as MNIST [24]. [22] proposes the integration of gradient-based GMMs with a continual learning framework by using GMMs as classifiers. In contrast to the above methods, we are the first to leverage the natural advantages of GMMs to preserve learned knowledge while effectively learning new knowledge for CISS tasks.

### 2.3 Class Incremental Semantic Segmentation

Class Incremental Semantic Segmentation (CISS) is first introduced by [30], which utilizes standard knowledge distillation to alleviate catastrophic forgetting. [4] addresses the unique issue of background shift by designing an unbiased function. Since then, various methods have focused on the aforementioned main challenges in CISS. Distillation-based methods develop the varied distillation terms [38, 1, 44, 55] to maintain old class knowledge. The pseudo-label strategy [13, 3, 50, 32] is proven to effectively mitigate background shift. Exampler-based methods employ memory selection mechanism [61] or generative model [28] to replay old class samples in incremental steps. Some methods employ additional information, such as saliency maps [5] and mask proposals [52, 53], to reduce semantic ambiguity in background regions.

Additionally, prototype-based replay methods [7, 60] leverage stored prototypes, where each old class is represented by a single prototype, to generate pseudo-features of old classes. Although achieving remarkable performance, the use of a single prototype for a class may not be sufficient to capture the complete distribution of that class, because the real-world data often exhibit multimodal characteristics. Meanwhile, the former [7] ignores the issue of mismatch between unchanged prototypes and anisotropic distribution of old classes during incremental processes. The latter [60] alleviates this problem by updating the centroids of the prototypes, but still suffers from limited robustness due to the unchanged variances [37]. In contrast, our method continuously estimates the precise distribution of each old class using a GMM with natural multiple attributes, and adapts to anisotropic distributions through dynamic GMM updates, striking an advantage.

## 3 Method

### 3.1 Problem Definition

In the class incremental semantic segmentation (CISS) task, a model is expected to learn classes over multiple steps, and we assume that there are $T$ steps. At each step $t$, the model $\mathcal{M}^t$ comprises a feature extractor $f^t$ and a classifier $g^t$. The dataset $\mathcal{D}^t$ for training the model consists a set of pairs $(\boldsymbol{x}^t, \boldsymbol{y}^t)$, where $\boldsymbol{x}^t$ is a input image and $\boldsymbol{y}^t$ is the corresponding segmentation label. Noting that $\boldsymbol{y}^t$ only exhibits the labels in current classes $\mathcal{C}^t$, while all other class (i.e. learned classes $\mathcal{C}^{1:t-1}$ and

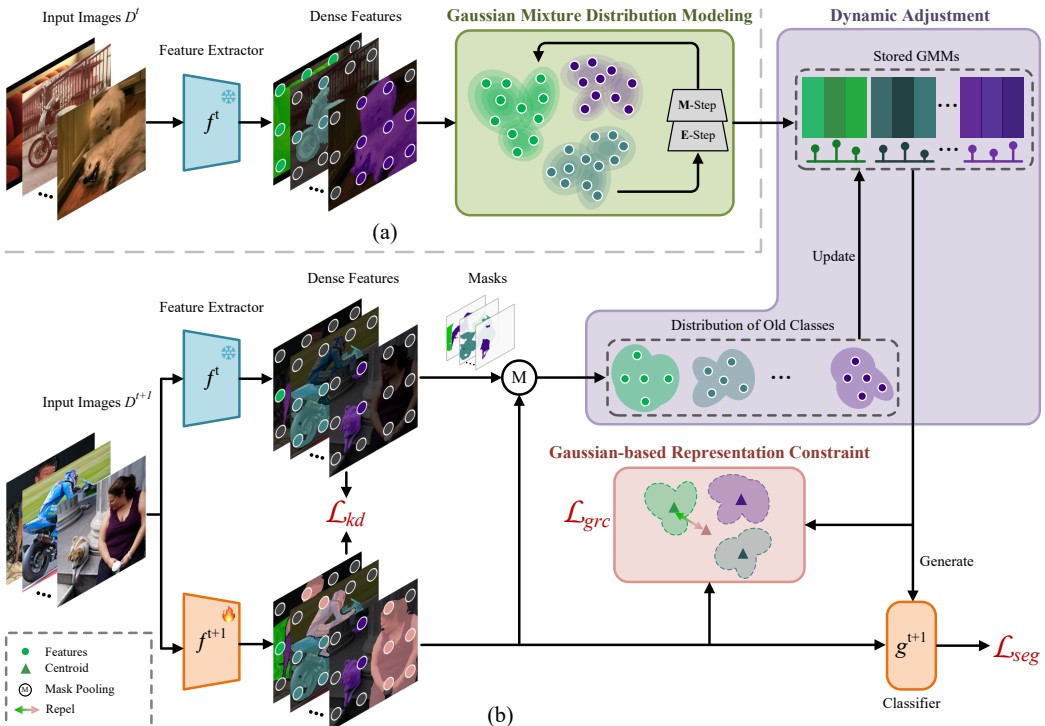

Figure 1: Overview of our proposed CoGaMiD. (a) Given the dataset $\mathcal{D}^t$ at step $t$, after conventional training, we estimate the distribution of classes at this step using the corresponding GMMs, which are optimized by the EM algorithm. (b) The features of old classes, obtained by the current model using old class masks, is applied in the Dynamic Adjustment (DA) strategy to update parameters of the stored GMMs. Meanwhile, a Gaussian-based Representation Constraint (GRC) loss is proposed to maintain the discriminative distance between new classes and old ones.

future classes $\mathcal{C}^{t+1:T}$) are labeled as background $\mathcal{C}_b$. Once the incremental step $t$ is completed, the model is required to perform segmentation for all seen classes $\mathcal{C}^{1:t}$.

## 3.2 Overview of the Proposed Method

We present an overview of our proposed CoGaMiD in Fig. 1. After training at step $t$, we freeze the feature extractor $f^t$ and estimate the distribution of each class in the current set of classes $\mathcal{C}^t$ using independent multivariate Gaussians based on the training dataset $\mathcal{D}^t$. The estimated distributions are then stored as GMMs, denoted as $\mathcal{G}^t$. In the next step $t + 1$, given a batch of data, both the previous frozen feature extractor $f^t$ and the current trainable feature extractor $f^{t+1}$ are used simultaneously to obtain dense feature representations. Following [5, 6, 52], the knowledge distillation loss is adopted as a fundamental supervision to preserve old class knowledge in the feature space. The pseudo-features of old classes, generated by stored GMMs (i.e., $\mathcal{G}^1, \ldots, \mathcal{G}^t$), combined with the output features of $f^{t+1}$, serve as input to the classifier $g^{t+1}$. The centroids of the new classes are obtained by masked average pooling of the output from $f^{t+1}$. These centroids are then used to compute the representation distances between them and the geometric centroids of the stored GMMs, forming the Gaussian-based Representation Constraint (GRC) loss. Additionally, we update the stored GMMs using the old class features extracted from the current model at intervals of several training epochs. We refer to this process as the Dynamic Adjustment (DA) strategy, which enables the stored GMMs to adapt to the anisotropic representations of old classes caused by parameter updates in the current model.

## 3.3 Gaussian Mixture Distribution Modeling and Generation

As previously stated, a single prototype in existing CISS methods [6, 60] is inadequate for capturing the complete distribution of each class, particularly when the feature space is inherently multimodal.

In this situation, the efficiency of the pseudo-features replay strategy is limited. Therefore, we propose to assign a GMM for each learned foreground class and store those GMMs for generating pseudo-features in subsequent steps.

Specifically, after training at each step $t$, the feature extractor $f^t$ is froze and utilized to extract foreground features from the training dataset $\mathcal{D}^t$. Subsequently, Gaussian Mixture Models (GMMs) are employed to estimate the distribution of the current classes $\mathcal{C}^t$ in the feature space. It is assumed that the features distribution of each class $c$ follows a mixture distribution consisting of $K$ Gaussian components. We denote the $k$-th mixture component is $\mathcal{N}(\boldsymbol{\mu}_k, \boldsymbol{\Sigma}_k)$, where $\boldsymbol{\mu}_k$ is a D-dimensional mean vector while $\boldsymbol{\Sigma}_k$ is a $D \times D$ covariance matrix. The feature distribution of each class $c$ can be modeled by linear combination of $K$ mixture components:

$$p(\boldsymbol{x}|c; \boldsymbol{\phi}_c) = \sum_{k=1}^{K} p(k|c; \boldsymbol{\pi}_c) p(\boldsymbol{x}|c, k; \boldsymbol{\mu}_c, \boldsymbol{\Sigma}_c) = \sum_{k=1}^{K} \pi_{ck} \mathcal{N}(\boldsymbol{x}; \boldsymbol{\mu}_{ck}, \boldsymbol{\Sigma}_{ck}), \tag{1}$$

where $p(k|c; \boldsymbol{\pi}_c)$ is the prior probability belonging to $k$-th Gaussians, i.e., $\sum_k \pi_{ck} = 1$, and $\phi_c = \{\boldsymbol{\pi}_c, \boldsymbol{\mu}_c, \boldsymbol{\Sigma}_c\}$ represents the parameters of GMMs. The multimodal nature of GMMs enables the accurate estimation of feature distributions, thereby providing a significant advantage over prototypes that assume unimodality for each class.

Our goal is to find the optimal parameters of the GMMs for each class, i.e., $\{\phi_c^*\}_{c=1}^{\mathcal{C}^t}$, by maximizing the log likelihood over the features of this class in the current training dataset $\mathcal{D}^t$:

$$\boldsymbol{\phi}_c^* = \arg\max_{\boldsymbol{\phi}_c} \sum_{i=1}^{N_c} \log \sum_{k=1}^{K} p(\boldsymbol{x}_i, k|c; \boldsymbol{\phi}_c), \tag{2}$$

where $N_c$ denotes the number of features for class $c$. We employ the EM (Expectation-Maximization) algorithm, which consists of two main steps: the E-step (Expectation step) and the M-step (Maximization step), to iteratively approximate the parameters of GMMs until convergence to a local optimum. In the E-step, given a feature sample $\boldsymbol{x}_i$, the posterior probability generated by $k$-th Gaussian component is computed as:

$$\gamma_{ik} = \frac{\pi_{ck} \cdot \mathcal{N}(\boldsymbol{x}_i \mid \boldsymbol{\mu}_{ck}, \boldsymbol{\Sigma}_{ck})}{\sum_{j=1}^{K} \pi_{cj} \cdot \mathcal{N}(\boldsymbol{x}_i \mid \boldsymbol{\mu}_{cj}, \boldsymbol{\Sigma}_{cj})}. \tag{3}$$

In the M-step, the parameters of $k$-th Gaussian component are updated based on the above probability:

$$\pi_{ck} = \frac{1}{N_c} \sum_{i=1}^{N_c} \gamma_{ik}, \quad \boldsymbol{\mu}_{ck} = \frac{\sum_{i=1}^{N_c} \gamma_{ik} \boldsymbol{x}_i}{\sum_{i=1}^{N_c} \gamma_{ik}}, \quad \boldsymbol{\Sigma}_{ck} = \frac{\sum_{i=1}^{N_c} \gamma_{ik} (\boldsymbol{x}_i - \boldsymbol{\mu}_{ck})(\boldsymbol{x}_i - \boldsymbol{\mu}_{ck})^T}{\sum_{i=1}^{N_c} \gamma_{ik}}. \tag{4}$$

Next, in order to generate the pseudo-features with stored GMMs in the next step $t$, we need to confirm the proportion of each class. For a foreground class $c$ in the current training dataset $\mathcal{D}^t$, which comprises $n$ image-label pairs $(\boldsymbol{x}^t, \boldsymbol{y}^t)$, we have:

$$N_c = \sum_{i=1}^{n} \sum_{j=1}^{h \times w} \delta\{\tilde{\boldsymbol{y}}_{i,j}^t = c\}. \tag{5}$$

Here $\delta\{\cdot\}$ is the indicator function, and $\tilde{\boldsymbol{y}}_i^t \in \mathbb{R}^{h \times w}$ denotes the downsampled label aligned with the spatial size of features. With the stored parameters of GMMs, $\{\phi_c^*\}_{c=1}^{\mathcal{C}^{1:t}}$, and the statistics, we can generate the pseudo-features of each class for reproducing in the next step by: $\mathcal{F}_c = \mathcal{S}(\phi_c^*, N_c)$, where $\mathcal{S}(\cdot)$ denotes the Gaussian sampling function. The obtained pseudo-features of old classes are uniformly assigned in each training iteration, enabling the classifier to capture discriminative characteristics between new and learned classes in the feature space.

### 3.4 Dynamic Adjustment Strategy

Precise pseudo-features generated by the GMMs are essential for refining the decision boundary. They effectively provide negative samples to support the learning of the new classes. During the incremental training process, the model parameters should be updated to accommodate new classes.

**Algorithm 1:** Dynamic Adjustment Strategy

---

**Input:** Training dataset $\mathcal{D}^{t+1}$, extractors $f^t$ and $f^{t+1}$, stored GMMs $\{\phi_c^*\}_{c=1}^{\mathcal{C}^{1:t}}$, training interval $e$
**Output:** Updated GMMs $\{\hat{\phi}_c^*\}_{c=1}^{\mathcal{C}^{1:t}}$

1 **while** *the end of each interval $e$* **do**
2      Freeze extractors $f^t$ and $f^{t+1}$;
3      **for** $(\boldsymbol{x}^t, \boldsymbol{y}^t) \in D^{t+1}$ **do**
4          Compute the mask maps $m$ based on Eq. 6;
5          Obtain old class features $\boldsymbol{F}_c$ from extractor $f^{t+1}$ based on Eq. 7;
6      **end**
7      Initialize GMMs with $\{\phi_c^*\}_{c=1}^{\mathcal{C}^{1:t}}$;
8      **for** $c$ *in* $\mathcal{C}^{1:t}$ **do**
9          Sample pseudo-features $\boldsymbol{\mathcal{F}_c} = \mathcal{S}(\phi_c^*, N_c)$;
10          **while** *not converged* **do**
11              Estimate responsibility $\gamma_k = \frac{\pi_{ck} \cdot \mathcal{N}(\{\boldsymbol{\mathcal{F}}_c, \boldsymbol{F}_c\} | \boldsymbol{\mu}_{ck}, \boldsymbol{\Sigma}_{ck})}{\sum_{j=1}^{K} \pi_{cj} \cdot \mathcal{N}(\{\boldsymbol{\mathcal{F}}_c, \boldsymbol{F}_c\} | \boldsymbol{\mu}_{cj}, \boldsymbol{\Sigma}_{cj})}$ ;        /* The E-step
12              Update the parameters $\{\boldsymbol{\pi}_c, \boldsymbol{\mu}_c, \boldsymbol{\Sigma}_c\}$ by Eq. 4; ;        /* The M-step
13          **end**
14          $\hat{\phi}_c^* \leftarrow \{\boldsymbol{\pi}_c, \boldsymbol{\mu}_c, \boldsymbol{\Sigma}_c\}$;
15      **end**
16 **end**

---

In this situation, the features of old classes, obtained from the trainable feature extractor, may exhibit anisotropy [16, 37], wherein the features shift in various directions. This anisotropy results in a mismatch between the fixed distributions obtained by stored GMMs and the continually evolving distributions of old classes. Consequently, a Dynamic Adjustment (DA) strategy is developed to address this issue.

At intervals of each $e$ training epochs in step $t+1$, we keep the extractor $f^{t+1}$ frozen to obtain the features of old classes $\mathcal{C}^{1:t}$ from the current training dataset $\mathcal{D}^{t+1}$. Firstly, we obtain the masks of old classes by combining predictions of the old model, $p^t$, with the current ground truth label $\boldsymbol{y}^{t+1}$:

$$m = \begin{cases} p^t, & (p^t \in \mathcal{C}^{1:t}) \wedge (\boldsymbol{y}^{t+1} \in c_b) \\ 0, & otherwise \end{cases}, \tag{6}$$

where $c_b$ denotes the background class. Then, given the $n$ input images in $\mathcal{D}^{t+1}$, we can obtain the features of the old classes based on above masks:

$$\boldsymbol{F}_c = \{F_1^{t+1} \times \delta\{\tilde{m}_1 = c\}, \ldots, F_n^{t+1} \times \delta\{\tilde{m}_n = c\}\}. \tag{7}$$

Here, $F_n^{t+1}$ denotes the features extracted by $f^{t+1}$ from the $n$-th input image, and $\tilde{m}$ represents the downsampled mask aligned with the spatial size of the features. We believe that the old class features extracted from the current model intuitively contain the anisotropic distribution, which cannot be accurately reproduced by the stored GMMs. Therefore, we utilize them to seasonably update the stored GMMs using EM algorithm. With continuously updated GMMs, we are able to generate matched pseudo-features that immediately adapt to the anisotropic distribution of old classes, facilitating the learning of new classes. The main procedure is summarized in algorithm 1.

### 3.5 Gaussian-based Representation Constraint

Above continual Gaussian mixture distribution modeling method addresses the issue of overfitting to new classes in classifiers. However, the feature extractor may generate ambiguous features for new and old classes with similar semantics. This ambiguity leads to an overlapping decision boundary, confusing the learning of classifiers. Additionally, the ambiguous features of the new class also adversely affect the subsequent Gaussian mixture modeling. Our goal is to maintain a sufficiently discriminative distance between similar classes in the latent space. Based on the above consideration, we introduce a Gaussian-based Representation Constraint (GRC) loss.

Table 1: Quantitative comparison on Pascal VOC 2012 between our method and previous CNN-based methods (top half) and Transformer-based methods (bottom half) under the *overlapped* setting. † means results from our re-implementation. ‡ represents the results are from [53].

| Method | VOC 15-1 (6 steps) | | | VOC 5-3 (6 steps) | | | VOC 10-1 (11 steps) | | | VOC 2-2 (10 steps) | | | VOC 1-1 (20 steps) | | |
| --- | --- | --- | --- | --- | --- | --- | --- | --- | --- | --- | --- | --- | --- | --- | --- |
| | 0-15 | 16-20 | all | 0-5 | 6-20 | all | 0-10 | 11-20 | all | 0-2 | 3-20 | all | 0-1 | 2-20 | all |
| MiB† [4] | 35.1 | 13.5 | 29.7 | 57.1 | 42.6 | 46.7 | 12.3 | 13.1 | 12.7 | 41.7 | 26.0 | 28.2 | 38.5 | 8.1 | 11.0 |
| PLOP† [13] | 65.1 | 21.1 | 54.6 | 41.1 | 23.4 | 25.9 | 44.0 | 15.5 | 30.5 | 24.1 | 11.9 | 13.7 | 12.4 | 11.9 | 4.7 |
| GSC [10] | 72.1 | 24.4 | 60.8 | 32.7 | 30.1 | 30.9 | 50.6 | 17.3 | 34.7 | - | - | - | - | - | - |
| SSUL† [5] | 77.3 | 36.6 | 67.6 | 72.4 | 50.7 | 56.9 | 71.3 | 46.0 | 59.3 | 62.4 | 42.5 | 45.3 | 52.6 | 27.5 | 29.9 |
| DKD† [1] | 78.1 | 42.7 | 69.7 | 69.6 | 53.5 | 58.1 | 73.1 | 46.5 | 60.4 | 60.5 | 45.8 | 47.9 | 56.1 | 24.6 | 27.6 |
| EWF [46] | 77.7 | 32.7 | 67.0 | 61.7 | 42.2 | 47.7 | 71.5 | 30.3 | 51.9 | - | - | - | - | - | - |
| RCIL [51] | 70.6 | 23.7 | 59.4 | 65.3 | 41.5 | 50.3 | 55.4 | 15.1 | 34.3 | 28.3 | 19.0 | 19.4 | - | - | - |
| IDEC [55] | 77.0 | 36.5 | 67.3 | 67.1 | 49.0 | 54.1 | 70.7 | 46.3 | 59.1 | - | - | - | - | - | - |
| CS²K [11] | 77.9 | 46.4 | 70.4 | 58.4 | 53.4 | 54.8 | 74.4 | 47.2 | 61.5 | - | - | - | - | - | - |
| STAR† [6] | 79.5 | 50.6 | 72.6 | 71.9 | 61.5 | 64.4 | 73.1 | 55.4 | 64.7 | 59.2 | 55.0 | 55.6 | 43.6 | 35.7 | 36.5 |
| Ours | 80.1 | 53.6 | 73.8 | 73.7 | 63.1 | 66.1 | 73.9 | 57.0 | 65.8 | 62.8 | 58.9 | 59.4 | 61.6 | 43.3 | 45.0 |
| Joint (CNN)‡ | 82.7 | 75.0 | 80.9 | 81.4 | 80.7 | 80.9 | 82.1 | 79.6 | 80.9 | 76.5 | 81.6 | 80.9 | 93.2 | 79.6 | 80.9 |
| MiB† [4] | 35.0 | 43.2 | 36.9 | 55.2 | 48.9 | 50.7 | 11.4 | 18.9 | 15.0 | 41.1 | 29.3 | 31.0 | 40.3 | 10.2 | 13.1 |
| SSUL† [5] | 78.1 | 33.4 | 67.5 | 72.8 | 51.2 | 57.4 | 74.3 | 51.0 | 63.2 | 60.3 | 40.6 | 44.0 | 51.8 | 26.2 | 28.6 |
| MicroSeg† [52] | 80.5 | 40.8 | 71.0 | 77.8 | 60.3 | 65.3 | 73.5 | 53.0 | 63.8 | 64.8 | 43.4 | 46.5 | 70.4 | 35.6 | 38.9 |
| Incrementer [38] | 79.6 | 59.6 | 75.6 | - | - | - | 77.6 | 60.3 | 70.2 | - | - | - | - | - | - |
| NeST [47] | 76.8 | 57.2 | 72.2 | - | - | - | 64.3 | 28.3 | 47.2 | - | - | - | - | - | - |
| CoinSeg† [53] | 82.7 | 52.5 | 75.5 | 76.1 | 65.4 | 68.5 | 80.1 | 60.0 | 70.5 | 70.1 | 63.3 | 64.3 | 72.1 | 40.3 | 43.3 |
| STAR† [6] | 80.7 | 57.3 | 75.1 | 76.6 | 68.2 | 70.6 | 79.8 | 61.4 | 71.0 | 70.5 | 64.8 | 65.6 | 70.8 | 41.7 | 44.5 |
| Ours | 83.2 | 61.2 | 78.0 | 79.9 | 72.7 | 74.7 | 81.1 | 65.9 | 73.8 | 73.4 | 70.0 | 70.5 | 79.8 | 51.5 | 54.2 |
| Joint (TranS)‡ | 83.8 | 79.3 | 82.7 | 81.1 | 83.3 | 82.7 | 82.4 | 83.0 | 82.7 | 75.8 | 83.9 | 82.7 | 92.0 | 81.7 | 82.7 |

Inspired by contrastive learning, we believe that the feature centroids of new classes should be sufficiently separated from those of the most similar old classes. Specifically, at step $t+1$, given a batch $B$ of images from the training dataset $\mathcal{D}^{t+1}$, we compute the feature centroids of new classes $\mathcal{C}^{t+1}$ by:

$$\boldsymbol{\mu}_{n_c} = \frac{\sum_{i \in B} \sum_{j=1}^{h \times w} (F_{i,j}^{t+1} \times \delta\{\tilde{\boldsymbol{y}}_{i,j}^{t+1} = c\})}{\left\| \sum_{i \in B} \sum_{j=1}^{h \times w} (F_{i,j}^{t+1} \times \delta\{\tilde{\boldsymbol{y}}_{i,j}^{t+1} = c\}) \right\|_2}, \tag{8}$$

where $\|\cdot\|_2$ is the L2 normalization. The stored GMMs have modeled the complete distribution of old classes in previous steps. Thus, we approximately estimate the centroids of old classes using the stored GMMs. For each old class $c$, we compute a geometric centroid by weighted combing the corresponding multi-attribute Gaussian means:

$$\boldsymbol{\mu}_{o_c} = \frac{\sum_{k=1}^{K} (\pi_{ck} \cdot \boldsymbol{\mu}_{ck})}{\left\| \sum_{k=1}^{K} (\pi_{ck} \cdot \boldsymbol{\mu}_{ck}) \right\|_2}. \tag{9}$$

After computing the feature centroids for all old classes, we can achieve the GRC loss as:

$$\mathcal{L}_{grc} = \frac{1}{|\mathcal{C}^{t+1}|} \sum_{n_c \in \mathcal{C}^{t+1}} \frac{1}{\min_{o_c \in \mathcal{C}^{1:t}} \|\boldsymbol{\mu}_{n_c} - \boldsymbol{\mu}_{o_c}\|_2}. \tag{10}$$

With $\mathcal{L}_{GRC}$, a sufficient representational distance is maintained between new classes and their most similar old counterparts. Reciprocally, the discriminative decision boundaries enable precise modeling of the distribution of new classes using GMMs, thereby facilitating the learning in subsequent incremental steps. The total loss of our method is:

$$\mathcal{L} = \mathcal{L}_{seg} + \alpha \mathcal{L}_{kd} + \beta \mathcal{L}_{grc}. \tag{11}$$

Here $\mathcal{L}_{seg}$ consists of a multiple Binary Cross-Entropy (mBCE) loss and an uncertainty loss [52, 5, 60], and $\mathcal{L}_{kd}$ is the refined knowledge distillation loss [6]. Hyperparameters $\alpha$ and $\beta$ balance these terms.

## 4 Experiments

### 4.1 Experimental Setup

**Datasets and evaluation metric.** We conduct comprehensive experiments on two public datasets: Pascal VOC 2012 [15] and ADE20K [57]. Pascal VOC 2012 contains 10,582 training images and

Table 2: Quantitative comparison on ADE20K between our method and previous CNN-based methods (top half) and Transformer-based methods (bottom half) under the *overlapped* setting. [†] means results from our re-implementation. [‡] represents the results are from [53].

| Method | ADE 100-50 (2 steps) | | | ADE 50-50 (3 steps) | | | ADE 100-10 (6 steps) | | | ADE 100-5 (11 steps) | | |
|---|---|---|---|---|---|---|---|---|---|---|---|---|
| | 0-100 | 101-150 | all | 0-50 | 51-150 | all | 0-100 | 101-150 | all | 0-100 | 101-150 | all |
| MiB [4] | 40.5 | 17.2 | 32.8 | 45.6 | 21.0 | 29.3 | 38.2 | 11.1 | 29.2 | 36.0 | 5.7 | 26.0 |
| PLOP[†] [13] | 41.9 | 14.9 | 32.9 | 48.8 | 21.0 | 30.4 | 40.5 | 13.6 | 31.6 | 39.1 | 7.8 | 28.8 |
| SDR [31] | 40.5 | 17.2 | 32.8 | 40.9 | 23.8 | 29.5 | 37.3 | 12.1 | 28.9 | 33.0 | 10.6 | 25.6 |
| SSUL [5] | 41.3 | 18.0 | 33.6 | 48.4 | 20.2 | 29.6 | 40.2 | 18.8 | 33.1 | 39.9 | 17.4 | 32.5 |
| IDEC [55] | 42.0 | 18.2 | 34.1 | 47.4 | 26.0 | 33.1 | 40.3 | 17.6 | 32.7 | 39.2 | 14.6 | 31.0 |
| STAR[†] [6] | 42.4 | 24.2 | 36.4 | 48.7 | 27.2 | 34.4 | 42.0 | 20.6 | 34.9 | 41.7 | 17.5 | 33.7 |
| NeST [47] | 42.3 | 22.8 | 35.8 | 48.2 | 27.4 | 34.4 | 40.7 | 19.0 | 33.5 | 39.4 | 15.5 | 31.5 |
| Ours | **43.1** | **24.7** | **37.0** | **49.3** | **27.8** | **35.1** | **42.5** | **22.4** | **35.8** | **42.3** | **21.0** | **35.2** |
| Joint (CNN)[‡] | 43.8 | 28.9 | 38.9 | 51.1 | 33.3 | 38.9 | 43.8 | 28.9 | 38.9 | 43.8 | 28.9 | 38.9 |
| SSUL [5] | 41.9 | 20.1 | 34.6 | 49.5 | 21.3 | 30.7 | 40.7 | 19.0 | 33.5 | 41.3 | 16.0 | 32.9 |
| MicroSeg[†] [52] | 41.1 | 24.1 | 35.4 | 49.8 | 23.9 | 32.5 | 41.0 | 22.6 | 34.8 | 41.2 | 21.0 | 34.5 |
| STAR[†] [6] | 42.8 | 26.4 | 37.4 | 49.2 | 28.1 | 35.2 | 42.5 | 25.1 | 36.7 | 42.2 | 24.3 | 36.3 |
| NeST [47] | 43.5 | 26.5 | 37.9 | 49.7 | 29.3 | 36.2 | 41.8 | 23.8 | 35.9 | 40.5 | 19.9 | 33.7 |
| Ours | **43.9** | **27.3** | **38.4** | **49.9** | **29.8** | **36.6** | **43.7** | **26.5** | **38.0** | **43.6** | **25.8** | **37.7** |
| Joint (TranS)[‡] | 44.2 | 30.6 | 39.7 | 50.8 | 34.1 | 39.7 | 44.2 | 30.6 | 39.7 | 44.2 | 30.6 | 39.7 |

1449 validation images, encompassing 20 foreground classes. ADE20K includes 20,210 training images and 2,000 validation images, distributed across 150 classes. We follow previous works [4, 13] to use mean Intersection-over-Union (mIoU) as evaluation metric.

**Protocols.** We follow the protocols in [4] to evaluate our model across various incremental scenarios defined as as $N_b - N_n$, where $N_b$ and $N_n$ denote numbers of base and novel classes, respectively. For instance, in 15-1 scenario, training begins with 15 classes, followed by the addition of one new class at each incremental step. In this paper, we focus on more challenging long-term incremental scenarios, e.g., Pascal VOC 2012 1-1, 2-2, and ADE20K 100-5. CISS has two incremental settings: *disjoint* and *overlapped*, we mainly evaluate our approach under the *overlapped* setup in this paper, which is more challenging and realistic. More details for protocols are provided in Appendix A.1.

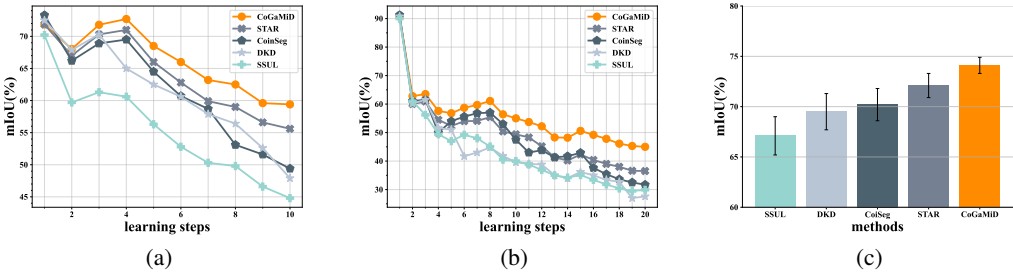

Figure 2: Visualization comparison for mIoU. Illustration of the change of mIoU with learning steps on Pascal VOC 2012 (a) 2-2 and (b) 1-1. (c) Average performance of 20 different incremental orders

**Implementation details.** Following previous works [5, 52, 6], we use a Deeplab-V3 [8] segmentation network with a ResNet-101 [18] backbone. We use SGD optimizer to optimize the network. The learning rate for the initial step is set to 0.001 and 0.00025 for Pascal VOC 2012 and ADE20K, respectively, and is reduced by a factor of 0.1 for the incremental steps. We train the network for 60 epochs on Pascal VOC and 100 epochs on ADE20K with 0.9 momentum and 0.0001 weight decay in all steps. The batch size is set to 24 for both datasets. For the hyper-parameters , $\alpha$, $\beta$ and $K$ are set to 5, 0.05, and 3, respectively. We conduct experiments on four NVIDIA RTX 4090 GPUs using PyTorch. More implementation details can be found in Appendix A.2.

## 4.2 Experimental Results

**Experiments on Pascal VOC 2012.** Besides the widely evaluated scenarios 15-1, 5-3, 10-1 in previous works, we verified the performance of the proposed method on two more challenging

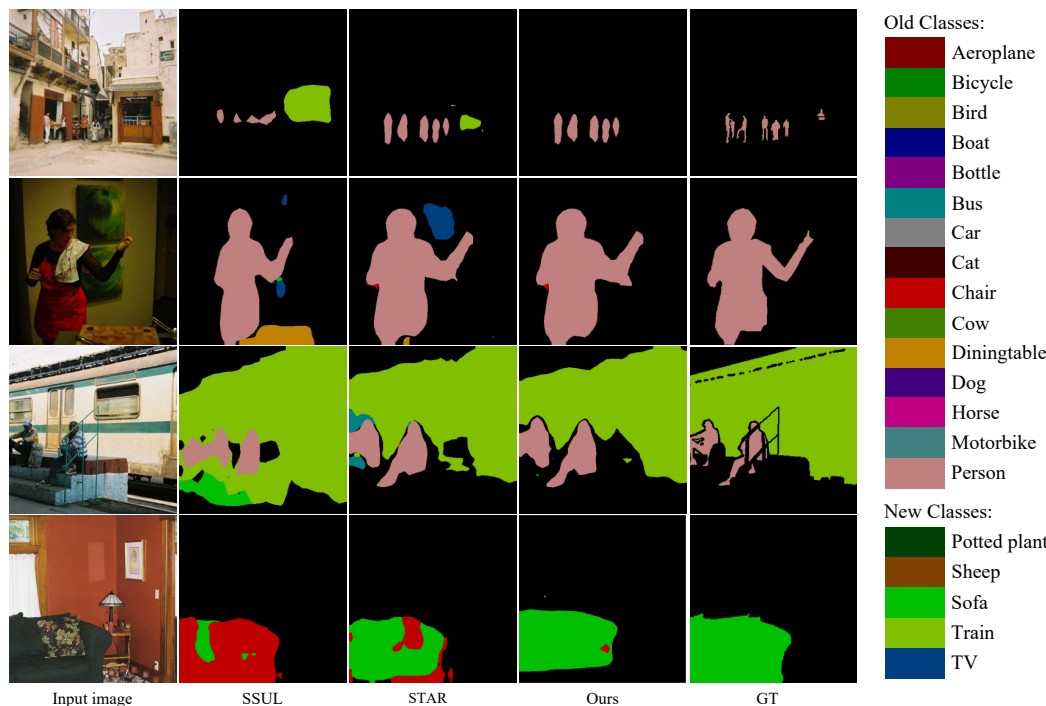

| Input image | SSUL | STAR | Ours | GT |

**Old Classes:**
- Aeroplane
- Bicycle
- Bird
- Boat
- Bottle
- Bus
- Car
- Cat
- Chair
- Cow
- Diningtable
- Dog
- Horse
- Motorbike
- Person

**New Classes:**
- Potted plant
- Sheep
- Sofa
- Train
- TV

Figure 3: Qualitative comparison on Pascal VOC 2012 between ours and previous methods.

incremental scenarios, e.g., 2-2 (10 steps), 1-1 (20 steps). Such long-term incremental scenarios are more meaningful and more closely aligned with realistic applications. As shown in Tab. 14,we present a quantitative comparison between our method and state-of-the-art CISS approaches [6, 38, 47, 53]. To effectively highlight the performance of our approach, we also present results from joint training, which serve as upper bounds. For ResNet-based models, our method demonstrates superior performance over existing methods across all experimental scenarios. For instance, on more challenging scenarios 2-2 and 1-1, which contain fewer classes in the initial step and learn novel classes from more steps, our method surpasses others by 3.8 and 8.5 mIoU, respectively. Meanwhile, results with Swin-B show that the proposed method is suitable for Transformer-based models. Leveraging the superior representation capabilities of Swin-B, our method achieves an improvement of nearly 10 mIoU for all classes on the most challenging 1-1 scenario.

Furthermore, Fig. 2a and Fig. 2b show the change of average mIoU for all seen classes under the scenarios of 2-2 and 1-1 at the corresponding learning steps, respectively. For instance, at 2-2 step 5, the average mIoU of seen classes 0-10 will be calculated. We observe that our method achieves similar performance with other methods at the initial step. However, in subsequent incremental steps, other methods exhibit a significant performance degradation, while our method effectively slows down the drop. This suggests that CoGaMiD causes less forgetting of previously learned knowledge, even on more challenging long-term scenarios. We also show the average performance with 20 different incremental orders on 15-1 scenario in Fig. 2c. Compared to previous works, our method demonstrates better robustness with a lower standard deviation, affirming its adaptability to diverse incremental learning settings. In Fig. 3, we visualize the qualitative results for the images from VOC 15-1. We observe that SSUL partially maintains performance in the old classes, e.g., *person*; however, it tends to overfit the novel classes, e.g., *train* and *TV*. Moreover, SSUL frequently produces confused predictions, reducing the performance of several classes (see *chair* and *sofa* in the second column). Although STAR alleviates this phenomenon to some extent, a similar problem still exists. Obviously, our method effectively learns the new classes while preserving knowledge of learned classes with high stability (third column). More qualitative and quantitative results are reported in Appendix A.4.

**Experiments on ADE20K.** For the more challenging ADE20K benchmarks, we present the experimental results across various scenarios in Table 2. In the short-term scenarios 100-50 and 50-50, our method achieves an improvement of at least 0.6 mIoU compared to other approaches. In the

Table 3: Ablation study of components in the proposed method on VOC 15-1.

| Baseline | GMMs | DA | GRC | VOC 15-1 (6 steps) 0-15 | 16-20 | all |
|:---:|:---:|:---:|:---:|:---:|:---:|:---:|
| ✓ | | | | 77.8 | 30.9 | 66.6 |
| ✓ | ✓ | | | 79.1 | 50.5 | 72.3 |
| ✓ | ✓ | ✓ | | 79.9 | 51.8 | 73.2 |
| ✓ | ✓ | ✓ | ✓ | **80.1** | **53.6** | **73.8** |

Table 4: Ablation study for Gaussian mixtures $K$, coefficients of loss $\alpha$, and $\beta$ on VOC 15-1.

| $K$ | VOC 15-1 (6 steps) 0-15 | 16-20 | all | $\alpha$ | VOC 15-1 (6 steps) 0-15 | 16-20 | all | $\beta$ | VOC 15-1 (6 steps) 0-15 | 16-20 | all |
|:---:|:---:|:---:|:---:|:---:|:---:|:---:|:---:|:---:|:---:|:---:|:---:|
| 3 | **80.1** | **53.6** | **73.8** | 3 | 79.8 | 53.0 | 73.4 | 0.01 | 79.5 | 51.9 | 72.9 |
| 5 | 79.6 | 53.1 | 73.3 | 5 | **80.1** | **53.6** | **73.8** | 0.05 | **80.1** | **53.6** | **73.8** |
| 7 | 79.8 | 53.3 | 73.5 | 7 | 80.0 | 52.7 | 73.5 | 0.1 | 79.9 | 52.8 | 73.3 |

most challenging long-term scenario, 100-5, our method surpasses the state-of-the-art approach by a substantial margin of 1.5 mIoU. Using Swin-B as the backbone further enhances the performance of our method. For instance, in 100-5 scenario, our method achieves 37.7 mIoU, which represents an increase of 2.5 mIoU compared to our ResNet-based method, and it surpasses the best method with same backbone by 1.4 mIoU. Qualitative results are show in Appendix A.4.

### 4.3 Ablation Study

**Ablation study on proposed components of CoGaMiD.** Tab. 3 shows the contributions of three components of our approach on Pascal VOC 2012, including Gaussian mixture distribution models (GMMs), dynamic adjustment (DA) strategy, and Gaussian-based representation constraint (GRC) loss. The first row refers to the baseline with $\mathcal{L}_{seg}$ and $\mathcal{L}_{kd}$. Modeling Gaussian mixture distribution of learned classes and generating pseudo-features increases both stability and plasticity, which are the most significant factors in incremental learning. With the DA strategy, benefiting from the adaptive alignment between updated GMMs and the anisotropic distribution, the performance is further refined. As for GRC loss, from the results in the third and last rows of the table, the application of GRC decreases the influence of confusion between new classes and similar old ones, significantly improving the performance of new classes. Combination of all components achieve the best performance.

**Gaussian mixtures.** Tab. 4 left shows the influence of Gaussian mixtures. From the table, we observe that increasing the number of Gaussian mixtures results in a slight decrease in performance. We believe that the distribution of most classes can be effectively estimated using a moderate number of Gaussian mixtures, while employing a larger number of mixtures may lead to overfitting the training dataset distribution. Thus, we empirically choose $K = 3$ for all experiments.

**Coefficients.** Tab. 4 right illustrates the influence of coefficients: $\alpha$ and $\beta$. The results show that, in most cases, our method is not highly sensitive to the coefficients. Given the constraints imposed by spatial limitations in the main text, further ablation studies are included in Appendix A.3.

## 5 Conclusions

In this paper, we propose an effective method, CoGaMiD, aiming at CISS problems by continually modeling distribution of the learned classes via GMMs. We first introduce the GMMs to estimate the multivariate distribution of each old class in the corresponding steps and store them to generate pseudo-features in subsequent steps. In order adapt to the anisotropic features that arise during the model training, we design a dynamic adjustment strategy to update stored GMMs using the features of old classes in the current data streams. Furthermore, we develop a Gaussian-based representation constraint loss to maintain the discriminative distance between new classes and similar old ones. Experiments demonstrate the effectiveness of our method, especially in long-term incremental scenarios, outperforming previous state-of-the-art CISS methods.

## 6 Acknowledgment

We would like to thank the HPC Platform of Huazhong University of Science and Technology for providing computational resources.

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

# A Appendix

## A.1 More Protocols

Previous works [4, 13, 28, 34] consider CISS for two incremental settings: *disjoint* and *overlapped*. In the *disjoint* setting, for each incremental step $t$, the training images only contain the pixels belonging to the classes that have been learned and will learned in the current step, i.e., $\mathcal{C}^{1:t}$. In the *overlapped* setting, future classes, $\mathcal{C}^{t+1:T}$ may appear in the current step but are labeled as background classes. Thus, we primarily report the results for the *overlapped* setting in the main body, while the results for the *disjoint* setting will be presented in the subsequent appendix.

## A.2 More Implementation Details

Following [53, 47], we also choose Swin-B [26] pretrained on ImageNet-1K [23] as the backbone to implement our method. For the segmentation head, we employ the dual-head architecture from CoinSeg [53], which includes a dense prediction head and a proposal classification branch. We follow [53, 52] to generate 100 class-agnostic proposals using a parameter-fixed Mask2Former [9] pre-trained on MS-COCO, which serves as supervision for the proposal classification branch. We use AdamW as the optimizer and train the network with a learning rate of 0.0001 for all steps. We train the network for 50 epochs on Pascal VOC 2012 and ADE20K with the batch size is set 16 and 8, respectively.

## A.3 More Ablation Study

**Proposed components.** We show extra ablations of components in Tab. 5. For the second row, we train the network without pseudo-features generated by stored GMMs and do not perform the dynamic adjustment (DA) strategy. Lacking the precise boundary of the old class, the models overfit to the novel classes, and the GRC loss fails to capture the robust features of the novel classes in the current step. This confusion is further exacerbated in the subsequent steps, since the incorrect distribution of novel classes is estimated and stored, thereby leading to the performance decrease. Using GMMs enables GRC to turn disadvantages into advantages (see last row in Tab. 5 and second row in Tab. 3). Based on above consideration, we believe that the effectiveness of GRC loss should be built on the precise representation of each learned class. Without DA strategy, the performance of our method decreases by 0.7 mIoU for all classes. This further demonstrates the effectiveness of the idea that continual adapt GMMs to the evolving models.

Table 5: Extra ablation study of components in the proposed method on Pascal VOC 2012 15-1.

| Baseline | GMMs | DA | GRC | VOC 15-1 (6 steps) | | |
| --- | --- | --- | --- | --- | --- | --- |
| | | | | 0-15 | 16-20 | all |
| ✓ | | | | 77.8 | 30.9 | 66.6 |
| ✓ | | | ✓ | 77.0 | 29.8 | 65.8 |
| ✓ | ✓ | | ✓ | 79.7 | 52.1 | 73.1 |

**GMMs.** We show influence of different settings on GMMs estimation in Tab 6. As for the GMMs parameters, i.e., $\mu_c$, we employ three type of initialization: 'random', 'k-means', 'feature means', where 'feature means' denotes that we initialize the $K$ Gaussian mixtures for each class with the corresponding the same feature means. The results show that 'k-means' initialization is preferable to other two. On the other hand, a large EM step number does not necessarily yield better performance. Therefore, we choose 500 as the maximum step number for the EM algorithm. In addition, to provide more results for K>3, we have conducted the experiments for more values of K, ranging from 4 to 7, as shown in Tab 7. The results reveal a slight decline on performance when K>3. We speculate that using a large number of Gaussian components may lead to partial overfitting, particularly for classes with simple feature distributions in the training set. We believe that K=3 is a relatively reasonable setting, holding performance and computational efficiency simultaneously. Moreover, we show results for different values of K on VOC 15-1, ranging from 1 to 3, for both the baseline and final model in Tab 11. The results suggest that using multiple Gaussians simultaneously improves the performance of the baseline and our final model, which demonstrates the superiority and necessity of

Table 6: Influence of GMMs settings on VOC 15-1.

| Initialization | VOC 15-1 (6 steps) | | | EM Steps | VOC 15-1 (6 steps) | | |
|---|---|---|---|---|---|---|---|
| | 0-15 | 16-20 | all | | 0-15 | 16-20 | all |
| random | 79.8 | 52.9 | 73.4 | 100 | 79.9 | 53.2 | 73.5 |
| k-means | **80.1** | **53.6** | **73.8** | 500 | 80.1 | **53.6** | **73.8** |
| feature means | 80.0 | 53.3 | 73.6 | 1000 | **80.2** | 52.6 | 73.6 |

Table 7: more values of K on VOC 15-1.

| K | VOC 15-1 (6 steps) | | |
|---|---|---|---|
| | 0-15 | 16-20 | all |
| 4 | 79.9 | 53.2 | 73.6 |
| 5 | 79.6 | 53.1 | 73.3 |
| 6 | 79.6 | 53.5 | 73.4 |
| 7 | 79.8 | 53.3 | 73.5 |

Table 8: Influence of different KD on VOC 15-1.

| Methods | VOC 15-1 (6 steps) | | |
|---|---|---|---|
| | 0-15 | 16-20 | all |
| Baseline (with standard KD from ILT [30]) | 77.6 | 20.5 | 64.0 |
| Baseline (with Local-POD from PLOP [13]) | 77.5 | 24.2 | 64.8 |
| Baseline (with OCFM from STAR [6]) | 77.8 | 30.9 | 66.6 |

Table 9: Experiments on computational and memory cost with previous exemplar-based methods.

| Methods | mIOU | GFLOPS | #Params | storage |
|---|---|---|---|---|
| SSUL [5] | 71.4 | 211 | 60M | 10M |
| STAR [6] | 72.6 | 272 | 59M | 0.04M |
| Ours | 73.8 | 278 | 59M | 0.12M |

our Gaussian Mixture distribution modeling approach compared to the use of a single prototype. We visualize the distributions of each base class in Fig. 4. From the figure, we observe that their shapes are often complex and irregular, making them difficult to approximate with a single Gaussian. We believe that the t-SNE results shown in Fig. 7 also partially reflect the multimodality of the most classes (e.g. 'person', 'sofa', 'cow', 'train', etc.). These class distributions can be more accurately modeled by Gaussian Mixture Models (GMMs) rather than a single Gaussian. Above qualitative and quantitative results further justify the necessity of adopting GMMs.

**Knowledge distillation.** we trained our baseline using OCFM loss in STAR as the form of. Since our focus is not on improving the knowledge distillation techniques that widely used in CISS, we hope to adopt the most effective distillation loss from existing methods to establish a stable baseline. We provide the results for the baseline using different distillation losses on VOC 15-1 in Tab 8. The results demonstrate the effectiveness of OCFM, which is why we adopted it as the knowledge distillation component and used it to train the model as our baseline.

**Computational and memory cost.** We provide a comparison of overall performance (mIoU) on VOC 15-1, computational complexity (GFLOPs), model size (#Params), and storage cost for exemplars or prototypes across the three methods, as shown in Tab 9. The results show that our method achieves performance gains of 1.2% at similar model sizes and GFLOPS. In addition, we use 0.12M memory for storaging GMMs, which is far lower than the SSUL that storage raw images. Even when compared with the prototype-based method, our storage cost only increase 0.08M that almost be ignored.

**Architectures.** Although our current research primarily focuses on Deeplab-based continual learing architectures, we attempt to implement our method on such a structure used in MBS [32] and conducted the experiments on VOC 15-1 compared with other methods, as exhibited in Tab 10.

Table 10: Experiments on Mask2Former-based architecture for VOC 15-1.

| Methods | VOC 15-1 (6 steps) | | |
|---|---|---|---|
| | 0-15 | 16-20 | all |
| MiB [4] | 72.6 | 23.1 | 61.7 |
| RBC [56] | 75.9 | 40.2 | 68.2 |
| INC [38] | 79.6 | 59.6 | 75.6 |
| MBS [32] | 82.6 | 72.2 | 80.6 |
| Ours | 82.9 | 73.9 | 81.2 |

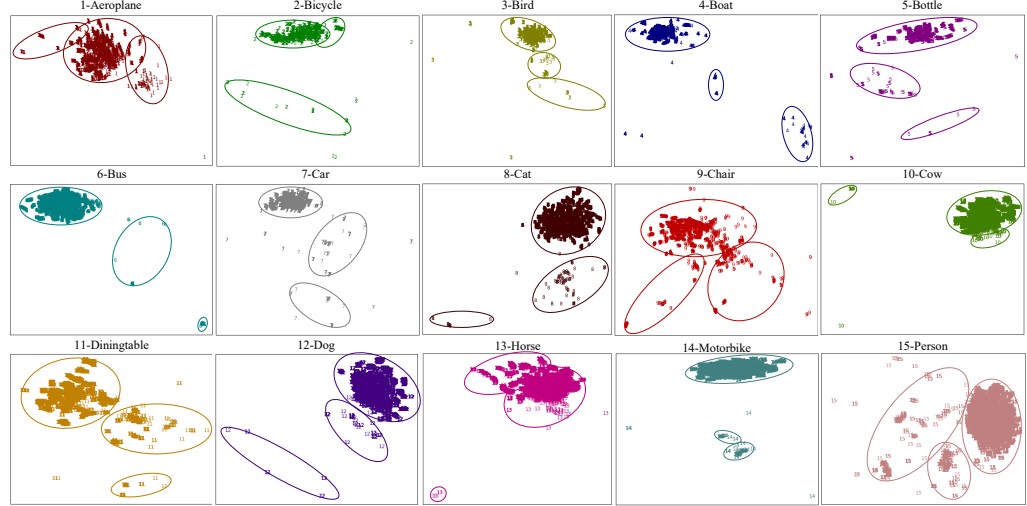

Figure 4: TSNE visualization for every base classes on VOC 15-1.

Table 11: Influence of K for both baseline and final model on VOC 15-1.

| Methods | VOC 15-1 (6 steps) | | |
|---|---|---|---|
| | 0-15 | 16-20 | all |
| Baseline | 77.8 | 30.9 | 66.6 |
| Baseline (K=1) | 78.4 | 38.6 | 68.9 |
| Baseline (K=2) | 78.9 | 47.8 | 71.5 |
| Baseline (K=3) | 78.4 | 38.6 | 68.9 |
| Ours (K=1) | 79.4 | 51.8 | 72.8 |
| Ours (K=2) | 80.0 | 52.7 | 73.5 |
| Ours (K=3) | 80.1 | 53.6 | 73.8 |

Table 12: Influence of $\mathcal{L}_{grc}$ settings on VOC 15-1.

| Centroids | VOC 15-1 (6 steps) | | |
|---|---|---|---|
| | 0-15 | 16-20 | all |
| original | 79.9 | 52.7 | 73.4 |
| geometric | **80.1** | **53.6** | **73.8** |

The results demonstrate the effectiveness of our method on such a structure, achieving competitive performance. We also believe that with further exploration, our method has the potential to perform even better on this structure.

**Centroids of old classes in $\mathcal{L}_{grc}$.** We consider the two types of old centroids in $\mathcal{L}_{grc}$: 'original' and 'geometric'. The type of 'original' represents the $K$ means derived from the stored GMMs

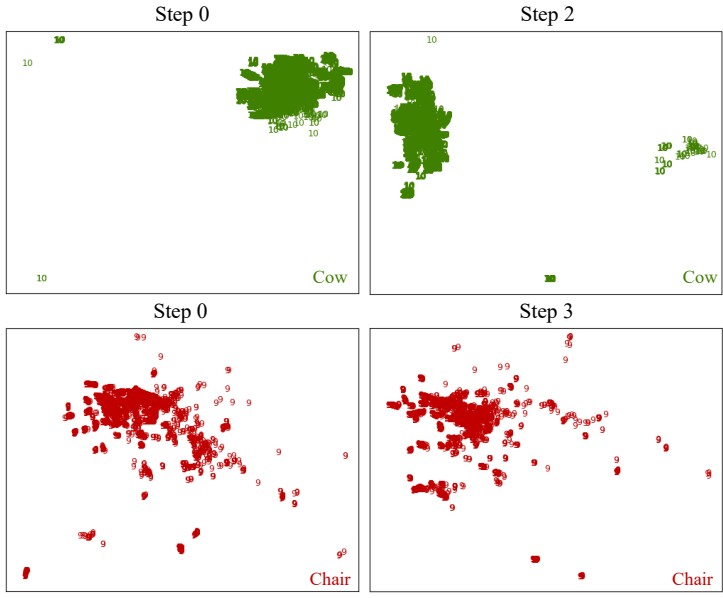

Figure 5: TSNE visualization for the class 'cow' and 'chair' at different learning step on VOC 15-1.

of each learned class are used to compute the L2 distance between these means and new class means, while 'geometric' denotes the geometric centroids that combined by $k$ means with the corresponding weights of $\pi_c$. Table 12 illustrates the impact of the aforementioned settings, indicating that the geometric approach yields better performance. We believe that the reason of the suboptimal performance obtained by 'original' centroids may be due to treating every Gaussian centroid equally, or overemphasizing the small probability distribution with few feature samples. In this case, the learning space of the new classes will be compressed, acquiring insufficient representations.

**Combination with memory sampling strategy.** We follow the works [5, 52, 6] to employ the memory sampling strategy to further enhance the performance of our method. Tab. 13 shows the results of our method with different memory numbers. With 20 additional raw images providing more realistic features, our method achieves performance that is nearly comparable to the previous best memory-based approach. The performance is further improved with more memory, while the required storage is half of that needed by other methods.

Table 13: Method with different memory number on VOC 15-1.

| Methods | VOC 15-1 (6 steps) | | |
| --- | --- | --- | --- |
| | 0-15 | 16-20 | all |
| Ours | 80.1 | 53.6 | 73.8 |
| Ours-M(20) | 80.1 | 55.2 | 74.2 |
| Ours-M(50) | **80.2** | **57.5** | **74.8** |
| SSUL-M(100) | 78.4 | 49.0 | 71.4 |
| DKD-M(100) | 78.8 | 52.4 | 72.5 |
| STAR-M(100) | 79.9 | 56.2 | 74.3 |

**Features anisotropy.** The anisotropy problem has been analyzed in the CIL[16, 37], which means the features shift in various directions during incremental learning process. We also observe the analogous phenomenon in the CISS. As depicted in Fig. 5, we find that the old class 'chair' and 'cow' showed more noticeable deviations in the subsequent incremental learning process, especially when the model learned the class like 'sofa' (step 3) and 'sheep' (step 2). We think that this is because the original feature space of the old class may cause confusion between similar new classes and old classes (e.g. 'chair' and 'sofa', 'cow' and 'sheep'). Therefore, the model tends to redistribute the features of these old classes in order to leave appropriate space for better learning of new classes.

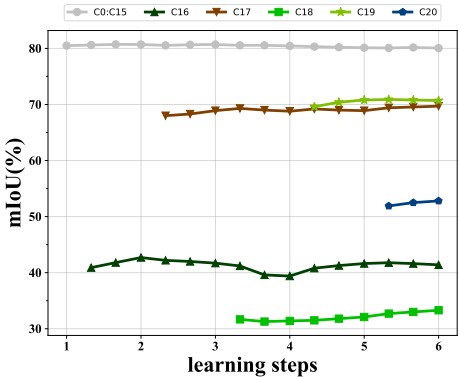

Figure 6: Illustration of the change of mIoU for old classes and each novel class on VOC 15-1.

Consequently, we propose a dynamic adjustment strategy that continuously updates the stored GMMs to mitigate this issue. The third row in Table 3 illustrates the effectiveness of the proposed strategy.

**Capability of preventing forgetting.** To further demonstrate the effectiveness of our method in preventing forgetting, we present the mIoU changes for the previously learned class and each novel class during incremental learning on VOC 15-1, as shown in Fig. 6. We report the mIoU of these classes after each 20 epochs training. We observe that the performance of the old classes remains nearly consistent throughout the incremental learning steps, benefiting from the proposed continual Gaussian mixture modeling method. For novel classes, our method enables the model to maintain performance in subsequent learning steps, even further improving the initial performance. We analyze that this is because when previously unknown categories are marked as new categories, the discriminant distance constraint reduces the confusion between the previous categories and the unknown categories in the background. These results demonstrate that the effectiveness of the proposed method in resisting forgetting.

## A.4 More Experimental Results

**Quantitative comparison under the disjoint setting.** In Tab. 14, we present a quantitative comparison between our STAR and previous methods under the *disjoint* setup. For CNN-based comparisons, our method achieves a slight advantage over other methods in short-term incremental steps, i.e., 19-1 and 15-5. This advantage becomes more pronounced on VOC 15-1. Similar results can be found in the Transformer-backbone methods, demonstrating that our method can handle various incremental settings with different backbone.

Table 14: Quantitative comparison on Pascal VOC 2012 between our method and previous CNN-based methods (top half) and Transformer-based methods (bottom half) under the *disjoint* setting.

| Method | VOC 19-1 (2 steps) | | | VOC 15-5 (2 steps) | | | VOC 15-1 (6 steps) | | |
|---|---|---|---|---|---|---|---|---|---|
| | 0-15 | 16-20 | all | 0-5 | 6-20 | all | 0-10 | 11-20 | all |
| MiB [4] | 69.6 | 25.6 | 67.4 | 71.8 | 43.3 | 64.7 | 46.2 | 12.9 | 37.9 |
| PLOP [13] | 75.4 | 38.9 | 73.6 | 71.0 | 42.8 | 64.3 | 57.9 | 13.7 | 46.5 |
| SSUL [5] | 77.4 | 22.4 | 74.8 | 76.4 | 45.6 | 69.1 | 74.0 | 32.2 | 64.0 |
| RBC [56] | 76.4 | 45.8 | 75.0 | 75.1 | 49.7 | 69.9 | 61.7 | 19.5 | 51.6 |
| MicroSeg [52] | 80.6 | 16.0 | 77.4 | 77.4 | 43.4 | 69.3 | 73.7 | 24.1 | 61.9 |
| CoinSeg [53] | 80.5 | 25.1 | 77.9 | 79.6 | 43.8 | 71.1 | 75.6 | 30.9 | 64.9 |
| STAR [6] | 77.9 | 43.4 | 76.2 | 78.4 | 57.4 | 73.4 | 78.1 | 46.6 | 70.6 |
| Ours | 79.8 | 46.4 | **78.2** | 78.9 | 58.2 | **74.0** | 78.9 | 49.2 | **71.8** |
| MiB [4] | 80.6 | 45.2 | 79.6 | 75.0 | 59.9 | 72.3 | 66.7 | 26.3 | 58.3 |
| CoinSeg [53] | 82.0 | 34.0 | 80.2 | 82.1 | 55.3 | 75.7 | 82.0 | 46.1 | 73.4 |
| Ours | 82.5 | 67.2 | **81.8** | 82.4 | 61.8 | **77.5** | 82.2 | 57.9 | **76.4** |

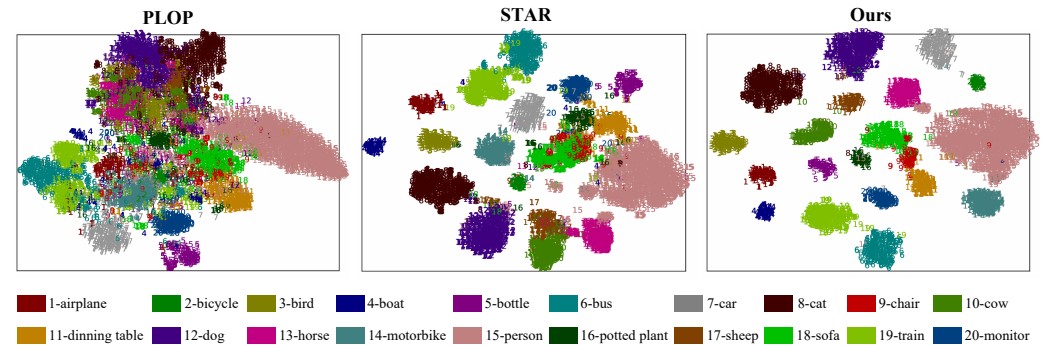

Figure 7: TSNE visualization on VOC 15-1. Numbers in the image represent the corresponding classes.

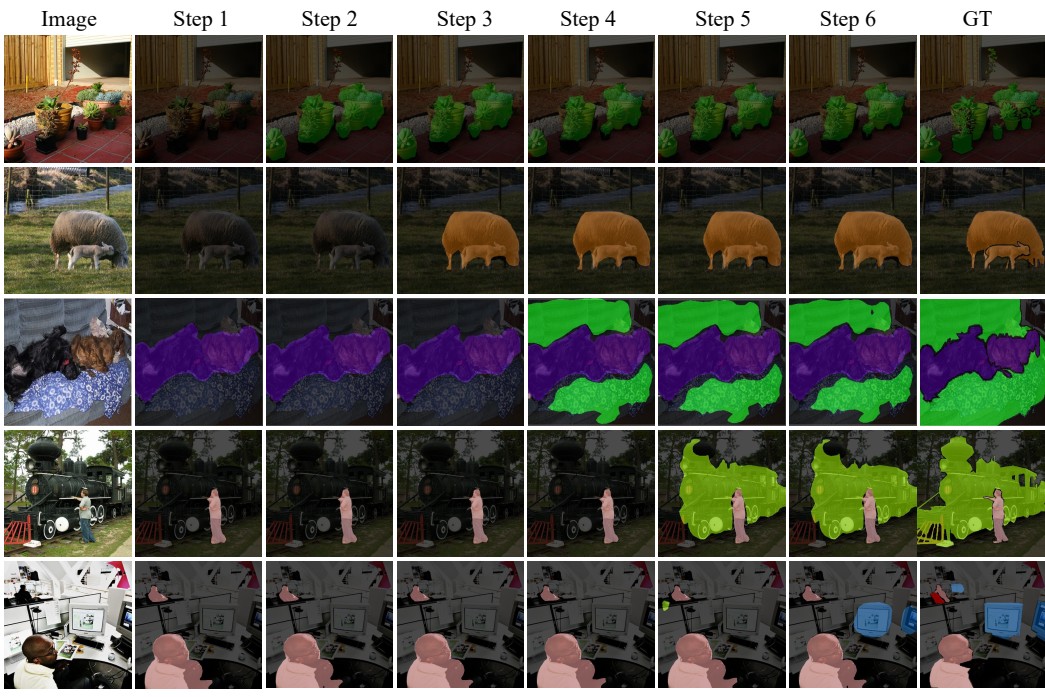

Figure 8: More qualitative results on VOC 15-1. *Plant*, *Sheep*, *Sofa*, *Train*, and *TV* are belong to the new classes.

**Qualitative comparison on VOC.** We present the t-SNE visualization of all classes for our method and previous methods on VOC 15-1, as depicted in Fig. 7. As illustrated in the figure, PLOP fails to learn discriminative representations during the incremental steps, resulting in significant forgetting. Benefiting from the feature replay strategy, STAR effectively preserves the knowledge of old classes. However, it introduces some confusion by neglecting the anisotropic distribution of old classes and imposing a fragile constraint on the distance between novel and old classes. The illustration demonstrates that our method achieves an excellent balance between plasticity and stability. This further supports the effectiveness of continual Gaussian mixture distribution modeling, which enables the model to exhibit strong anti-forgetting on old classes while also providing favorable adaptability to novel classes.

**More qualitative results.** In Fig. 8, we visualize the qualitative results of CoGaMiD for the same images after each learning step on VOC 15-1 (6 steps). For images containing only new classes, our method effectively learns these classes during the corresponding steps while maintaining consistent performance in subsequent steps. Regarding the sample set of *dog* and *sofa*, *dog* belongs to the old

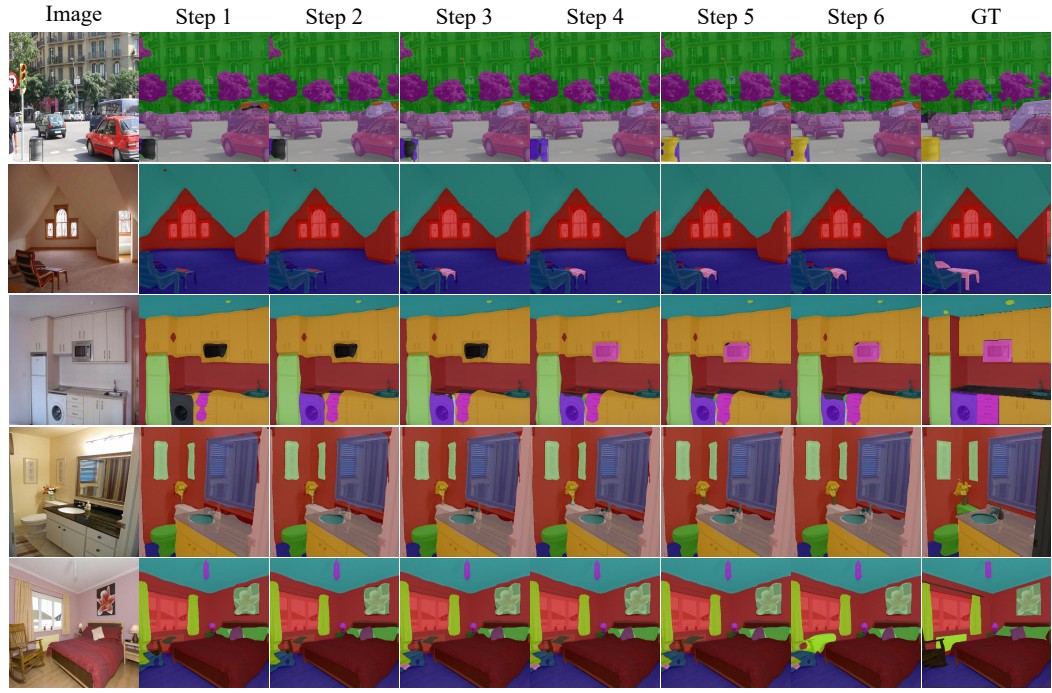

Figure 9: More qualitative results on ADE20K 100-10 (6 steps). *Stage* , *Stool* , *Washer* , *Vase* , and *Radiator* are belong to the new classes.

classes, while *sofa* is learned in step 4. Our method performs the complete segmentation of novel class *sofa* with the learned class *dog* is insusceptible, and similar phenomenon is shown in the fourth row. In the fifth row, a small portion of the background pixels is incorrectly classified as *train* due to slight overfitting during the learning of this class in step 5; however, this error is promptly corrected in the subsequent step. We present the qualitative results of our method for five images at corresponding steps on ADE20K 100-10 (6 steps), as illustrated in Fig. 9. The ADE20K dataset, with its extensive number of classes to be learned in both initial and incremental steps, poses particular challenges. As shown in these figures, our method effectively learns to predict new classes while simultaneously retaining the ability to identify old classes, demonstrating both stability and plasticity.

## A.5   Limitations and Future work

Through extensive experiments on two public benchmarks, our method shows excellent performance and achieves a favorable balance between stability and plasticity. However, estimating the distribution of classes using a predefined number of Gaussian mixtures may not be robust for all classes, given that the complexity of each class varies. Therefore, we will extend the method to an enhanced version that adaptively models the distribution with appropriate mixtures. Furthermore, we believe that compact representation in the initial step is crucial for the learning of subsequent steps. Thus, we will discuss that how to conduct the pre-constraint on the feature space of old classes in the initial step, reserving sufficient space for the learning of novel classes, in future work.

