# OpenReview forum: "Continual Gaussian Mixture Distribution Modeling for Class Incremental Semantic Segmentation"
_NeurIPS.cc/2025/Conference — NeurIPS 2025 poster_

### Official Review · Reviewer_vPPN · 2025-06-19

**Clarity:** 2
**Significance:** 3
**Originality:** 3
**Rating:** 4
**Confidence:** 3

**Summary:**

The paper integrates Gaussian mixture distribution for class-incremental segmentation, which tackles the challenge of performance degradation due to overfitting to new classes.
The paper proposes modeling the latent class feature distribution as a GMM. It also introduces a dynamic adjustment strategy that iteratively updates GMM parameters based on feature representations from the current model, and a Gaussian-based representation constraint loss that encourages separation between features of new and old classes.

**Questions:**

Please refer to the items listed in the weakness section. It would be helpful if the authors could provide an explanation to improve the clarity of the paper.

**Ethical Concerns:**

["NO or VERY MINOR ethics concerns only"]

**Final Justification:**

My initial rating assumed the listed issues could be resolved during the discussion phase. The authors' rebuttal has addressed most of my concerns. Considering other reviewers' comments, I intend to maintain my original (positive) rating.

**Limitations:**

Limitations are discussed at the end of the paper.
This paper does not introduce potential negative socialtal impact.

**Quality:**

3

**Strengths And Weaknesses:**

-- Strengths

The contrastive learning-inspired loss is a reasonable and effective way to preserve class separability during incremental learning, preventing overlap between similar old and new classes.

The dynamic adjustment strategy is a reasonable design to handle continual drift during incremental steps.

Experiment results on Pascal VOC and ADE20K datasets are promising, especially in challenging long-term incremental settings.


-- Weaknesses

Assuming latent features follow a fixed GMM with K=3 components per class is strong. While it may work empirically, it is recommended to at least provide more discussion.

Some mathematical notations (e.g., subscript/superscript usage) are not properly explained. It would improve the paper if the authors included a summary table of all symbols and notations.

It is not clearly explained how the segmentation network leverages or integrates the GMM features during classification. It is recommended to provide either diagrams or mathematical formulations specifying the corresponding inputs and outputs, and/or any special layer architecture if used.

---

> ### Author Rebuttal · Authors · 2025-07-25
>
> We greatly appreciate the reviewer's comprehensive review and recommendation of our work. Our responses to each point are as follows:
>
> ___
>
> **1. Empirically assuming fixed GMM components for per class in latent features.**
>
> Considering the generalizability of our method, we chose to use a fixed number of components (K=3) per class for all incremental scenarios, based on the results presented in Table 4 of the paper. We show additional results on VOC 15-1 using more values of K as follow:
>
> **K**|**0-15**|**16-20**|**all**
> :-|-|-|-|
> 1|79.4|51.8|72.8
> 2|80.0|52.7|73.5
> 3|80.1|53.6|73.8
> 4|79.9|53.2|73.6
> 5|79.6|53.1|73.3
> 6|79.6|53.5|73.4
> 7|79.8|53.3|73.5
>
> The results suggest that the overall performance of our method remains stable across most K values, indicating that it is not highly sensitive to the selection of this hyperparameter. Additionaly, the results reveal a slight decline in performance when the number of Gaussian components beyond three. We speculate that using a large number of Gaussian components may lead to partial overfitting, particularly for classes with simple feature distributions in the training set. Therefore, we choose K=3 for the final model, holding performance and computational efficiency simultaneously. Obviously, using a fixed number of GMM components is unlikely to be fully robust across all classes, because the complexity of feature space varies significantly between different classes, as discussed in Appendix  A.6 of our manuscript. We look forward to addressing this issue in future work by assigning an adaptive number of Gaussian components to each class, enabling the model to better adapt to varying levels of feature distribution complexity across different incremental scenarios. We will include these results and give more disscussion in the final version.
>
>
> ____
>
> **2. Mathematical notations.**
>
> We apologize for the unclear explaination of some mathematical notations in the manuscript. We will review these notations and provide detailed explanations in the final version. Following your valuable suggestion. Below is a summary table of all symbols with subscripts/superscripts, along with their corresponding notations, presented as comprehensively as possible. Please note that some vector symbols in the table below may be displayed unnormally due to platform limitations. We will present the standard symbols in the final version. In addition, some typos of formulas and symbols might result in misunderstanding (e.g. $\mu_{n_o}$  should be ${\mu}_{n_c}$ in Eq. 10.). We will modify them in the final version.
>
> **Symbols**|**Notations**|**Symbols**|**Notations**
> :-|-|-|-|
> $\mathcal{M}^t$|segmentation model at $t$ step|$\mathcal{D}^t$|training dataset at $t$ step
> $\mathcal{C}^t$|class set at $t$ step|$\mathcal{C}_b$|background class
> $f^t$|feature extractor at $t$ step|$g^t$|classifier at $t$ step
> $\mathcal{G}^{t}$|Gaussian Mixture Models (GMM) at $t$ step|$\boldsymbol{\mu}_{ck}$|mean vector of $k$-th mixture component in class $c$
> $\boldsymbol{\Sigma}_{ck}$|covariance matrix of $k$-th mixture component in class $c$|$\pi_{ck}$|probability of of $k$-th mixture component in class $c$
> $\tilde{\boldsymbol{y}}_{i}^{t}$|downsampled label of $i$-th image at $t$ step|$N_c$|number of pseudo-features to be generated for class $c$
> $\mathcal{F}_c$|pseudo-features for class $c$|$p^t$|predictions obtained from $t$-th step segmentation model $\mathcal{M}^t$
> $\tilde{m}_n$|downsampled mask of $n$-th image|$F_{c}$|real features for class $c$
> $\boldsymbol{\mu}_{o_c}$|geometric centroid for features of old class $c$|$\boldsymbol{\mu}_{n_c}$|centroid for features of new class $c$
>
> ___
> **3. Usage of GMM features during classification.**
>
> We appreciate your thoughtful recommendations. At step $t+1$, we utilize the storaged GMMs to generate pseudo-features for the old classes. Specifically, for each old class $c$, we reproduce the $N_c$ pseudo-features by per training epoch: $\mathcal{F}\_c = \mathcal{S}(\phi_c^*, N_c)$, where $\phi\_c^{\*}$ denotes the parameters of GMMs and $\mathcal{S}(\cdot)$ represents Gaussian sampling function. Then, pseudo-features are concatenated with the real features $F$ extracted from the input training images by the feature extractor $f^{t+1}$, and jointly fed into the classifier $g\^{t+1}$. Consequently, we obtain the corresponding pseudo-prediction $p$. We compute a standard mBCE loss by: $$\mathcal{L}\_{mbce}=-\frac{1}{N}\sum_{j=1}^{N}\sum_{c \in   \mathcal{C}^{t+1}}w\cdot y_{j, c}\cdot\log p_{j, c}+(1-y_{j,c})\cdot\log(1-p_{j,c})$$ Here, $N$ is the number of reproduced pseudo-features, $j$ is the index, $\mathcal{C}^{t+1}$ is the set of foreground classes at $t+1$ step, and $w$ is the positive weight. $y_{j,c}$ is the ground-truth of the $j$-th features sample (0 means it does not belong to class $c$, 1 means it does). In this way, the classifier $g^{t+1}$ can capture discriminative characteristics between new and learned classes in the feature space, achieving a good balance between stability and plasticity. We will include the such mathematical formulations and description in the final version.

---

> ### Comment · Reviewer_vPPN · 2025-08-04
>
> I thank the authors for preparing the rebuttal.
>
> I have a follow-up question for #3:
> For each pixel location on image, only one class is positive among all $c\in C$. Therefore, $y_{i,c}=0$ may dominate $L_{mbce}$.
>
> How do you set the weighting parameter $w$?

---

> ### Author Response · Authors · 2025-08-05
> **Thank you for the response**
>
> We thank the reviewer for the thoughtful response.
> ___
> As you mentioned, when $y_{i,c}=0$, the value of the first term in $\mathcal{L}\_{mbce}$ will be $0$ and the classfier $g^{t+1}$ is only supervised by the second term. This reflects our expectation that the classifier $g^{t+1}$  should assign lower probability scores when pseudo-features of old classes are treated as negative samples of new classes, thereby ensuring a clear decision boundary between old and new classes. Since we use the same mBCE loss function to calculate the losses of all classifier outputs, the form of $\mathcal{L}\_{mbce}$ that presented in the initial response can be regarded as the pseudo-prediction loss part, which is not affected by the weight $w$. The weighting parameter $w$ mainly works during the learning of new class pixels in the real images of $\mathcal{D}^{t+1}$.  We show whole $\mathcal{L}\_{mbce}^{all}$ for pseudo-features and real pixels as follow:
> $$\mathcal{L}\_{mbce}^{all}=-\frac{1}{H\cdot W+N}\sum_{j=1}^{H\cdot W+N}\sum_{c \in   \mathcal{C}^{t+1}}(w\cdot y_{j, c}\cdot\log p_{j, c}+(1-y_{j,c})\cdot\log(1-p_{j,c}))$$
>
> where $H\cdot W$ indicates the spatial size of the input image and $j$ is the index of pseudo-features or real pixels. Considering the imbalance between the number of new class samples and that of other samples (real background, and pixels or pseudo-features from other classes, we set the parameter $w$ to balance the weight of the two terms.
>
> We sincerely thank you for precious time and valuable comments on our work agin.

---

### Official Review · Reviewer_zb2d · 2025-06-30

**Clarity:** 3
**Significance:** 4
**Originality:** 4
**Rating:** 5
**Confidence:** 5

**Summary:**

This paper proposes a Continual Gaussian Mixture Distribution (CoGaMiD) modeling method for class-incremental semantic segmentation task, which considers the distribution characteristics of class prototypes in Gaussian Mixture Models (GMMs) and updates the prototype distribution of old classes through the Dynamic Adjustment (DA) strategy to solve the problem of distribution shift of old-class prototypes caused by extractor updates. The author conducted extensive experiments on the Pascal VOC and ADE20K datasets to validate the effectiveness of the proposed method.

**Questions:**

See weaknesses. I will modify my rate based on the author's responses to the questions in the weaknesses.

**Ethical Concerns:**

["NO or VERY MINOR ethics concerns only"]

**Final Justification:**

The author's response addressed my concerns, and the method is effective.  I raise my rating to 'Accept'.

**Limitations:**

yes

**Quality:**

3

**Strengths And Weaknesses:**

Strengths:
1. The problems with the single-prototype-based approach mentioned by the author are reasonable, and the idea proposed by the author to store the distribution of class features is intuitive.
2. The author's observation on the problem of class feature shift caused by the update of the extractor is insightful.
3. Strong performance is achieved on several benchmarks.



Weaknesses:
1. Is the form of $ L_{grc} $ inconsistent with what the author stated in line 212? The optimization goal of $ L_{grc} $ is more like shortening the distance between new classes and their most similar old counterparts. And there might be some typos, e.g., $ \mu_{n_o} $ in Eq. 10.
2. Based on the calculation of the geometric centroid of the old class according to Eq. 9, how to ensure a sufficiently clear decision boundary if some classes are widely distributed in the feature space? Will this also cause overlap?
3. Is the method of dynamically adjusting the distribution of old classes also considered a form of forgetting?
4. It is better to clarify how the prototype is used in the continual learning process and the specific setting of the baseline in the ablation studies, which will be more friendly for readers to follow.

---

> ### Author Rebuttal · Authors · 2025-07-27
>
> We appreciate your detailed comments and suggestions. Our responses to each point are as follows:
>
> ___
> **1.Inconsistent statement and typos.**
>
> We apologize for the misunderstanding caused by our oversights regarding the writting of Eq. 10. The goal of $L_{grc}$ is to maintain the discriminative distance between new classes and their most similar old counterparts. We will correct all identified issues in the final version:
>
> **a.** We will change $\mu_{n_o}$ to $\mu_{n_c}$ and modify Eq.10 to $L_{grc}=\frac{1}{|\mathcal{C}^{t+1}|}\sum_{n_c\in\mathcal{C}^{t+1}}\frac{1}{\min_{o_c\in\mathcal{C}^{1:t}}\||\mu_{n_c}-\mu_{o_c}\||_2}$ for consistant statement.  **b.** We will change $\pi_j$ to $\pi\_{cj}$ in Eq. 3 to refer to the specific class $c$.   **c.** We will modify the font style of $\pi\_{ck}$/$\pi\_{cj}$ from vector to scalar across the paper, which denotes a probability value blong to $k$-th Gaussian component for class $c$.
>
> ___
> **2.How to ensure a sufficiently clear decision boundary if some classes are widely distributed in the feature space? Will this also cause overlap?**
>
> Thank you for your thoughtful comment. We believe that a well-trained feature extractor enables features of the same class to be distributed in a unified and compact latent space. Consequently, the distances among the K mixture components of a given class tend to be closer to each other than to those of different classes. We argue that computing the geometric centroid provides a more intuitive and comprehensive representation of old class feature distributions than relying on the means of individual mixtures. For new class features, supervised by $L_{grc}$, it is easier to maintain an appropriate distance outside the distributions of old classes rather than being closely embedded within them. Based on the above analysis, we believe that using the geometric centroid formulation effectively avoids the issue of overlap. The t-SNE visualization results of Figure 5 in our manuscript further intuitively demonstrate that our method can ensure a sufficiently clear decision boundary. Additionally, we report the results of different $L_{grc}$ settings regarding the formulation of old class centroids on VOC 15-1 below (as shown in Appendix Table 7), to further verify the effectiveness of our design. Here, 'original' represents that using the K mean vectors derived from the stored GMMs as centriods of each learned class to compute the L2 distance between these centriods and new counterparts.
>
>
> |Centroids|0-15|16-20|all
> :-|-|-|-|
> original| 79.9| 52.7|73.4
>  geometric|80.1| 53.6|73.8
>
> ___
> **3.Wheter dynamically adjusting the distribution of old classes also be considered a form of forgetting?**
>
> There might be a misunderstanding. In our view, the primary cause of forgetting is that the model fails to capture the complete distribution of previously learned classes due to lacking the past data during the incremental learning process. Our method of Gaussian mixture distribution modeling utilizes stored GMMs to reproduce the complete distribution of previously learned classes, which has significantly alleviated the issue of forgetting. However, the distribution of the old classes may change because the model parameters are updated to accommodate new classes. This leads to a mismatch between the fixed distribution obtained by the stored GMMs and the evolving distribution of previously learned classes reflected in the continually updated model. Thus, we propose a dynamic adjustment (DA) strategy to update the GMMs, allowing them to continuously adapt to the changes in the distribution of previously learned classes within the model. Based on the above analysis, we tend to argue that our method of dynamically adjusting the distribution of previously learned classes addresses the issue of distribution mismatch, rather than representing a form of forgetting. The results presented in the table below demonstrate the effectiveness of our DA strategy.
>
> baseline|GMMs|DA|GRC|0-15|16-20|all
> :-|-|-|-|-|-|-|
> &#10004;|&#10004;| |&#10004;|79.7|52.1|73.1
> &#10004;|&#10004;|&#10004;|&#10004;|80.1|53.6|73.8
>
> ___
> **4.Usage of prototype and specific setting of the baseline.**
>
> Usage of prototype: In continual learning process, single-prototype based methods utilize stored prototypes obtained from previous learning steps to generate old class features. This is typically done by repeatly adding random Gaussian variances to the mean vectors of old classes. The generated features are then combined with real features and fed into the classifier to facilitate the learning of new classes.
>
> Specific setting of the baseline: In the ablation studies, we selected the segmentation network implemented with DeepLabv3 and ResNet-101 pretrained on ImageNet as our baseline model, which was surpervised by standard a multiple Binary Cross-Entropy (mBCE) loss in the initial step. During the incremental learning process, we incorporate the knowledge distillation loss from STAR [1], along with an uncertainty loss from Adapter [2], as additional supervision.
>
> We will clarify the usage of prototype and specific setting of the baseline in the final version to help the readers to follow.
>
> ___
> [1] Saving 100x storage: Prototype replay for reconstructing training sample distribution in class-incremental semantic segmentation. In NIPS 2023.
>
> [2] Adaptive prototype replay for class incremental semantic segmentation. In AAAI 2025.

---

> > ### Comment · Reviewer_zb2d · 2025-08-06
> >
> > Thank you for the author's thoughtful and detailed response. I appreciate the efforts and experiments the author has made in the response, which has well addressed most of my questions.
> >
> > I still have some remaining concerns: If the intra-class feature distribution is compact in the latent space, are K mixture components still necessary? In the ablation experiment in Table 4, only the results of K=3, 5, and 7 are shown, and the performance gap is not obvious. Could the experimental results of K=1 be presented?
> >
> > I hope the above can help you improve your work. I will think carefully and give my final rating.

---

> > > ### Author Response · Authors · 2025-08-06
> > >
> > > We thank the reviewer for the thoughtful response.
> > >
> > > ___
> > > **Q1. If the intra-class feature distribution is compact in the latent space, are K mixture components still necessary?**
> > >
> > > We believe that the compactness and multimodality of features are not mutually exclusive. The former reflects the degree of feature aggregation in the latent space, while the latter indicates the complexity of the feature distribution. For example,  K Gaussian components, combined from different perspectives, can also form a compact class representation. Therefore, in most cases, multiple Gaussian components capture the feature distribution of a class more accurately. The comparison between single and multiple Gaussian components, in response to your next question, further supports this observation.
> > >
> > > **Q2. Could the experimental results of K=1 be presented?**
> > >
> > > We appreciate your thoughtful recommendations. According to the valuable suggestion of reviewers vdnJ and vPPN, we have conducted the related experiments for different values of K. We show these results as follow:
> > >
> > > **Method**|**0-15**|**16-20**|**all**
> > > :-|-|-|-|
> > > Baseline|77.8|30.9|66.6
> > > Baseline (K=1)|78.4|38.6|68.9
> > > Baseline (K=2)|78.9|47.8|71.5
> > > Baseline (K=3)|79.1|50.5|72.3
> > > Ours (K=1)|79.4|51.8|72.8
> > > Ours (K=2)|80.0|52.7|73.5
> > > Ours (K=3)|80.1|53.6|73.8
> > >
> > > The results suggest that using multiple Gaussian components simultaneously enhances the performance of both the baseline and our final model, demonstrating the superiority of our Gaussian Mixture distribution modeling approach over the use of a single Gaussian. Obviously, the fixed number of K has its limitations, because the complexity of feature space varies significantly between different classes. We will explore the better method to assign an adaptive number of Gaussian components to each class, enabling the model to better adapt to varying levels of feature distribution complexity across different incremental scenarios.
> > >
> > >
> > > We sincerely thank you for precious time and valuable comments on our work agin.

---

> > > > ### Comment · Reviewer_zb2d · 2025-08-06
> > > >
> > > > Thank you for the author's response, which has addressed my concerns. I will consider the above content as the basis for my final rating.

---

> > > > > ### Author Response · Authors · 2025-08-06
> > > > >
> > > > > We are glad that our response has addressed your concerns. Thank you once again for your thoughtful review and insightful comments, which we believe have significantly improved the work.

---

> ### Comment · Area_Chair_mgvj · 2025-08-05
>
> Dear Reviewer zb2d,
>
> The author-rebuttal phase is now underway, and the authors have provided additional clarifications and performance results in their rebuttal. Could you please take a moment to review their response and engage in the discussion? In particular, we’d appreciate your thoughts on whether their revisions adequately address your initial concerns. Thank you for your time and valuable contributions.
>
> Best, Your AC

---

### Official Review · Reviewer_dzLn · 2025-06-30

**Clarity:** 3
**Significance:** 3
**Originality:** 3
**Rating:** 5
**Confidence:** 5

**Summary:**

This paper introduces CoGaMiD, a novel method for Class Incremental Semantic Segmentation (CISS). The core idea is to address the limitations of recent prototype-based approaches that cannot represent the complete features of each category. The method models the feature distribution of each class with a Gaussian Mixture Model (GMM) instead of a single prototype. The additional Dynamic Adjustment overcomes the shifting feature distribution of previous classes. Gaussian-based Representation Constraint (GRC) Loss maintain discriminability between new and old classes. The experiments on Pascal VOC 2012 and ADE 20K demonstrate the effectiveness of the approach.

**Questions:**

- Is this method effective for Mask2Former-based  structures compared to [1][2][3]?
- Is there any visualization result (t-SNE or segmentation result) to show anisotropy in detail?

**Ethical Concerns:**

["NO or VERY MINOR ethics concerns only"]

**Final Justification:**

The responses have addressed most of my initial concerns.

**Limitations:**

The authors have provided a discussion of limitations in Appendix A.6.

**Paper Formatting Concerns:**

The paper formatting meets the requirements.

**Quality:**

3

**Strengths And Weaknesses:**

Strengths:

- The paper is well written and easy to understand.
- The experiments are solid and the performance is impressive.
- The application of the Gaussian Mixture Model (GMM) balances the quality of features and the external space of buffer.

Weaknesses:

- Why is the Joint result of CNN different from previous methods? The previous work [1][2] shows that the Joint result of Deeplab-V3 with ResNet-101 is 77.4.
- Limited intuitive experiment results showing the anisotropy problem.
- Lack of experiments on computational and memory cost with previous exemplar-based methods.

[1] Cermelli, Fabio, et al. "Modeling the background for incremental learning in semantic segmentation." Proceedings of the IEEE/CVF Conference on Computer Vision and Pattern Recognition. 2020.
[2] Park, Gilhan, et al. "Mitigating Background Shift in Class-Incremental Semantic Segmentation." European Conference on Computer Vision. Cham: Springer Nature Switzerland, 2024.

---

> ### Author Rebuttal · Authors · 2025-07-26
>
> We greatly appreciate the reviewer's insightful review of our work. Our responses to each point are as follows:
>
> ___
> **1.Different joint result of CNN compared with previous methods.**
>
> As you mentioned, some methods reported different joint results (e.g., 77.4 in MiB [1] and MBS [2]). In our manuscript, we shown the joint results of CNN directly cited from CoinSeg [3] for the consideration of revealing the performance gap between the current CISS methods and the best upper bound. Based on our observations, these methods differ from each other in experimental settings (e.g., learning rate, batch size, and training epochs). Additionally, the segmentation network in Coinseg is trained with the assistance of proposals generated by Mask2Former, further enhancing the performance of joint training. Thus, we reproduce as many results on VOC 15-1 and joint results with correspond official code as possible as follow:
>
> Method|0-15|16-20|all
> :-|-|-|-|
> MiB|35.5|13.1|30.2
> PLOP|64.1|21.7|54.0
> DKD|78.3|40.7|69.4
> CoinSeg|80.6|36.2|70.1
> STAR|79.5|50.6|72.6
> Ours|80.1|53.6|73.8
> Joint (MiB)|79.3|73.0|77.8
> Joint (PLOP)|79.5|73.6|78.1
> Joint (DKD)|80.1|72.6|78.3
> Joint (STAR)|80.2|72.6|78.4
> Joint (Ours)|80.4|72.4|78.5
> Joint (CoinSeg)|82.7|75.0|80.9
>
> These results show that our approach achieves the smallest performance gap (4.7%) to ours own upper bound. Moreover, our joint result is largely consistent with other methods, except for CoinSeg. Therefore, we argue that our evaluation provides a fair comparison environment, and the joint result of Coinseg serves as a reference for the best upper bound. We will provide more such results of other incremental scenarios in the final version.
>
>
> ___
>
> **2.Limited intuitive experiment results showing the anisotropy problem. (W2 & Q2)**
>
> We are grateful for the reviewer's insightful comments. We roughly treat the change in old class features during the learning of new classes as an anisotropy problem, which was analyzed in FeCAM[4]. Considering your recommendation, we conducted visualization experiments on the feature distributions of some old classes at incremental steps on VOC 15-1. We observed that the shapes of some feature distributions have undergone certain changes, indicating that part of the old class features have deviated from their initial positions during the model's updating process. We truly want to show these visualizations but there is no way to add new figures during the rebuttal process. Therefore, we will add such visualizations with more analysis in the final version so please understand it.
>
> ___
> **3.Experiments on computational and memory cost with previous exemplar-based methods.**
>
> As suggested, we provide a comparison of the overall performance (mIoU) on VOC 15-1, computational complexity (GFLOPs), model size (#Params), and storage cost for exemplars or prototypes across the three methods, as follows:
>
> **Methods**|**mIoU**|**GFLOPS**|**#Params**|**storage**
> :-|-|-|-|-|
> SSUL|71.4|211|60M|10M
> STAR|72.6|272|59M|0.04M
> Ours|73.8|278|59M|0.12M
>
> The results show that our method achieves performance gains of 1.2% at similar model sizes and GFLOPS. In addition, we use 0.12M memory for storaging GMMs, which is far lower than the SSUL that storage raw images. Even when compared with the prototype-based method (STAR), our storage cost only increase 0.08M that almost be ignored.
>
> ___
>
> **4.Effectiveness of method for Mask2Former-based structures.**
>
> We appreciate your thoughtful recommendations. Our current research primarily focuses on Deeplab-based continual learing architectures. The Mask2Former-based structures in MBS [2] differ from the DeepLab-based architecture used in our manuscript, requiring us to make several adjustments to adapt the method for implementation on such a framework. This might take a relatively long time. Thus, we look forward to exploring the effectiveness of our method for Mask2Former-based structures in future work, and we kindly ask for your understanding.
>
> ____
>
> [1] Modeling the background for incremental learning in semantic segmentation. In CVPR 2020.
>
> [2] Mitigating Background Shift in Class-Incremental Semantic Segmentation. In ECCV 2024.
>
> [3] CoinSeg: Contrast Inter- and Intra- Class Representations for Incremental Segmentation. In ICCV 2023.
>
> [4] FeCAM:Exploiting the Heterogeneity of Class Distributions in Exemplar-Free Continual Learning. In NIPS 2023.

---

> > ### Comment · Reviewer_dzLn · 2025-08-06
> >
> > Thanks for carefully addressing my concerns with detailed responses.
> > The explanation and clarification addressed most of my concerns.
> > However, two further questions:
> > 1. For the anisotropy problem, as you mentioned that "some feature distributions have undergone certain changes, indicating that part of the old class features have deviated from their initial positions," could you specify which old classes show more noticeable deviations?
> > 2. In the Mask2Former-based experiments, what does “preliminarily attempted” mean?

---

> > > ### Author Response · Authors · 2025-08-06
> > >
> > > We thank the reviewer for the thoughtful response.
> > > ___
> > > **Q1.Which old classes show more noticeable deviations?**
> > >
> > > Among the feature distributions of old classes learned in the initial step, we found that the old class 'chair' and 'cow' showed more noticeable deviations in the subsequent incremental learning process, especially when the model learned the class like 'sofa' and 'sheep'.  We think that this is because the original feature space of the old class may cause confusion between similar new classes and old classes (e.g. 'chair' and 'sofa', 'cow' and 'sheep'). Therefore, the model tends to redistribute the features of these old classes in order to leave appropriate space for better learning of new classes. We will provide such visualizational results and detailed discussion in the next version.
> > >
> > >
> > > **Q2.In the Mask2Former-based experiments, what does “preliminarily attempted” mean?**
> > >
> > > There are two intermediate features in the Mask2Former-based structures: one extracted from the encoder and the other from the decoder output. In our preliminary attempt, we employed the Gaussian mixture models to estimate the output features of the decoder and generate the corresponding pseudo-features for old classes. These were then used to train the model with the assistance of the proposed dynamic adjustment strategy and the Gaussian-based representation constraint loss. We will further explore the effectiveness of our method on this structure by attempting to model the output features of the encoder or by incorporating the distillation loss proposed in MBS.
> > >
> > >
> > >
> > > We sincerely thank you for precious time and valuable comments on our work agin.

---

> > > > ### Comment · Reviewer_dzLn · 2025-08-06
> > > >
> > > > Thank you for your detailed response.
> > > > Your replies to my additional questions are generally reasonable.
> > > > As a result, I am willing to increase my score in the final rating.

---

> ### Author Response · Authors · 2025-08-03
> **Official Comment by Authors**
>
> Thank you for your thorough review and insightful questions again. Regarding your Q2 ('Is this method effective for Mask2Former-based structures compared to [1][2][3]?'), we preliminarily attempted to implement our method on such a structure used in MBS [2] and conducted the experiments on VOC 15-1 compared with other methods as follow:
>
> Method|1-15|16-20|all
> :-|-|-|-|
> MiB|72.6|23.1|61.7
> RBC|75.9|40.2|68.2
> INC|79.6|59.6|75.6
> MBS|82.6|72.2|80.6
> Ours|82.9|73.9|81.2
>
> The results demonstrate the effectiveness of our method on such a structure, achieving competitive performance. We also believe that with further exploration, our method has the potential to perform even better on this structure.
>
>
> [1] Modeling the background for incremental learning in semantic segmentation. In CVPR 2020.
>
> [2] Mitigating Background Shift in Class-Incremental Semantic Segmentation. In ECCV 2024.
>
> [3] Incrementer: Transformer for class-incremental semantic segmentation with knowledge distillation focusing on old class. In CVPR, 2023.
>
> [4] Rbc: Rectifying the biased context in continual semantic segmentation.In ECCV, 2022.

---

> ### Comment · Area_Chair_mgvj · 2025-08-05
>
> Dear Reviewer dzLn,
>
> The author-rebuttal phase is now underway, and the authors have provided additional clarifications and performance results in their rebuttal. Could you please take a moment to review their response and engage in the discussion? In particular, we’d appreciate your thoughts on whether their revisions adequately address your initial concerns. Thank you for your time and valuable contributions.
>
> Best, Your AC

---

> ### Author Response · Authors · 2025-08-06
>
> Thank you very much for your valuable feedback, time, and attention. We sincerely appreciate your encouragement. Your feedback and engagement throughout the review and discussion process has greatly improved our paper, and we will be sure to incorporate these relevant results and clarifications into our final version.

---

### Official Review · Reviewer_vdnJ · 2025-07-02

**Clarity:** 2
**Significance:** 2
**Originality:** 2
**Rating:** 4
**Confidence:** 5

**Summary:**

The authors present an novel framework, dubbed CoGaMiD, for class-incremental semantic segmentation. CoGaMiD utilizes a Gaussian Mixture Model to generate features for previously learned classes that may be inaccessible or scarcely available during incremental learning phases. These generated features facilitate training a segmentation model and play a crucial role in alleviating the issue of forgetting prior knowledge of old classes. Additionally, the authors propose a dynamic adjustment strategy to refine these generated features with real features collected during the incremental steps. They also introduce a Gaussian-based representation constraint loss aimed at learning discriminative features for new classes.

**Questions:**

Please read the questions in the weaknesses section.

**Ethical Concerns:**

["NO or VERY MINOR ethics concerns only"]

**Final Justification:**

Through the discussion process, I came to agree with the authors on the empirical effectiveness of the proposed multiple Gaussian approach. Additionally, the authors mentioned plans to further support the necessity of using GMMs through additional analyses such as feature distribution visualizations.

I look forward to seeing these enhancements in the final version and, with that expectation, I am raising my rating to borderline accept.

**Limitations:**

yes

**Quality:**

3

**Strengths And Weaknesses:**

**Strengths**

- CoGaMiD uses GMMs to generate features for previously learned classes, assisting a segmentation model in mitigating forgetting.

**Weaknesses**

- The authors utilize Gaussian Mixture Models (GMMs) to generate features of previously learned classes for the purpose of updating a segmentation model. In recent work, STAR [6], features of old classes are generated from Gaussian-distributed random variables collected from earlier stages. The Gaussian-based Representation Constraint (GRC) bears similarity to the similarity-aware discriminative loss in STAR, as both aim to ensure the means of Gaussians for new classes are distinct from those of old classes. The specifics of the refined knowledge distillation loss $L_{kd}$ in Eq. 11 are unclear. Is this referring to the effectiveness of the Old-Class Features Maintaining (OCFM) loss in STAR? Given these considerations, CoGaMiD appears to be closely related to STAR, with the primary distinction being the use of a Gaussian mixture model as opposed to a single Gaussian model, which suggests that CoGaMiD might be seen as an extension of STAR. It would be beneficial to clearly delineate the differences between CoGaMiD and STAR. Additionally, to demonstrate the effectiveness of using multiple Gaussians, a key distinction from STAR, it would be valuable to present results for different values of K, ranging from 1 to 3, for both the baseline and the final models.
- The performance observed when training with all samples simultaneously, referred to as 'joint,' represents an indirect upper boundary for the incremental model. In Tables 1 and 2, the results for the 'joint' method appear to be higher than those for DKD [1], a recent approach in CISS which utilizes binary cross-entropy loss, similar to CoGaMiD. To ensure a fair comparison and to highlight the effectiveness of the proposed methods without modifications to the architecture or minor techniques, it would be advantageous to include the reproduced results of some baseline models in Tables 1 and 2

---

> ### Author Rebuttal · Authors · 2025-07-29
>
> We sincerely appreciate the reviewers' thorough evaluation and constructive feedback. The major comments are answered below.
> ____
>
> **1.Difference between our method and STAR.**
>
> As you mentioned, some prior works have employed prototypes to generate the features of learned classes, with a representative being STAR [1] that you referred to. However, there were some limitations and challenges in the previous methods, which have been discussed in our submitted manuscript. Our main contribution is to address these problems in prototype-replay based CISS methods. We highlight clear differences between ours method and STAR from three points as follow:
>
> - **Gaussian Mixture Models (GMMs) vs. single prototype:** Firstly, STAR utilizes a single prototype to represent the complete distribution of each old class, which may be insufficient when the incoming data stream for a class is inherently multimodal. Secondly, the method of obtaining prototypes by directly computing the mean vectors of old class features is not easily extendable from a single prototype to multiple prototypes. Different from STAR, to address the aforementioned limitations, we are the first to ingeniously incorporate the multimodal characteristics of GMMs into the CISS framework, enabling accurate modeling of old class distributions through iterative learning with the Expectation-Maximization (EM) algorithm. The results from the table in the response '3.Results for different values of K.' also demonstrates the effectiveness of our use of multiple Gaussians.
>
> - **Dynamic GMMs adjustment vs. fixed prototype replay:** During incremental learning steps, the model needs to update its parameters to accommodate new classes. Inevitably, some old class features that are sensitive to parameter changes deviate from their original positions. This is more likely to result in a mismatch between the fixed prototypes in STAR and the changed feature distributions of the old classes. To address this issue, we propose a Dynamic Adjustment (DA) strategy that continuously updates the parameters of stored GMMs using the real-time features of old classes at each learning step, thereby adapting to the changed representations of old class distributions. The results on VOC 15-1 (Table 3 in our manuscript) below suggest that our DA strategy further improve the performance of the model on both of new and old classes.
> Baseline|GMMs|DA|0-15|16-20|all
> :-|-|-|-|-|-|
> &#10004;|&#10004;||79.1|50.5|72.3
> &#10004;|&#10004;|&#10004;|79.9|51.8|73.2
>
> - **Gaussian-based representation constraint (GRC) loss vs. similarity-aware discriminative (SAD) loss:** As you mentioned, we adopt a contrastive loss formulation similar to that of SAD. However, SAD selects a single prototype as the centroid of each old class, which ignores the non-uniform density and multimodal characteristics of feature distributions. In contrast, we refine the computation of old class centroids by adaptively combining multiple Gaussian means, weighted by the respective probability of each mixture component. This refinement enables a more faithful representation of the old class distributions. To further illustrate the impact of our GRC loss, we present additional experimental results on VOC 15-1 below, comparing the final models trained with either SAD or GRC loss. The results demonstrate that our GRC loss provides more effective guidance for learning new classes and significantly improves performance.
> Method|0-15|16-20|all
> :-|-|-|-|
> With $L\_{sad}$|79.8|52.1|73.2
> With $L_{grc}$|80.1|53.6|73.8
>
>
> In summary, our method diverges from STAR both in features modeling (single prototype vs. multiple Gaussian mixture components) and features replaying (fixed prototype vs. Dynamic GMMs). Moreover, our modification of anchors of old class in GRC loss is line with the disign of our method, fundamentally distinguishes it from previous SAD loss in STAR. From the results of our extensive experiments and ablation study, it can be seen that our method of continual Gaussian mixture distribution modeling (CoGaMiD) is effective, particularly in challenging multi-step scenarios (e.g. VOC 1-1 (20 steps) and ADE 100-5 (11 steps)).
>
> ___
> **2.The specifics of the refined knowledge distillation loss $L_{kd}$ in Eq. 11 are unclear. Is this referring to the effectiveness of the Old-Class Features Maintaining (OCFM) loss in STAR?**
>
> We apologize for unclearly notation of $L_{kd}$ in Eq. 11. As you guessed, we trained our baseline using OCFM loss in STAR as the form of $L_{kd}$. Since our focus is not on improving the knowledge distillation techniques that widely used in CISS, we hope to adopt the most effective distillation loss from existing methods to establish a stable baseline. We provide the results for baseline using different distillation losses on VOC 15-1 as follow:
>
> Method|0-15|16-20|all
> :-|-|-|-|
> Baseline (with standard KD from ILT [2])|77.6|20.5|64.0
> Baseline (with Local-POD from PLOP [3])|77.5|24.2|64.8
> Baseline (with OCFM from STAR [1])|77.8|30.9|66.6
>
> The results demonstrate the effectiveness of OCFM, which is why we adopted it as the knowledge distillation component and used it to train the model as our baseline. We will clarify the notation of $L_{kd}$ in the final version.
>
> ___
> **3.Results for different values of K.**
>
> Following your suggestion, we show results for different values of K on VOC 15-1, ranging from 1 to 3, for both the baseline and final models as follows:
>
> Method|0-15|16-20|all
> :-|-|-|-|
> Baseline|77.8|30.9|66.6
> Baseline (K=1)|78.4|38.6|68.9
> Baseline (K=2)|78.9|47.8|71.5
> Baseline (K=3)|79.1|50.5|72.3
> Ours (K=1)|79.4|51.8|72.8
> Ours (K=2)|80.0|52.7|73.5
> Ours (K=3)|80.1|53.6|73.8
>
> The results suggest that using multiple Gaussians simultaneously improves the performance of the baseline and our final model, which demonstrates the superiority of our Gaussian Mixture distribution modeling approach compared to the use of a single prototype.
>
> ___
> **4.The results for the 'joint' method appear to be higher than those for DKD, and present the reproduced results of some baseline models.**
>
> As you mentioned, some methods reported different joint results (e.g., 77.6 reported in DKD [4]). In our manuscript, we reported the joint results of CNN directly cited from CoinSeg [5], in order to highlight the performance gap between existing CISS methods and the best upper bound. Based on our observations, the differences in the joint results of these methods mainly stem from variations in the experimental settings (e.g., learning rate, batch size, and training epochs). Additionally, the segmentation network in CoinSeg is trained with the assistance of proposals generated by Mask2Former [5], which further enhances the performance of joint training. Following your suggestion, we conducted joint training using our method and reproduced as many results with the corresponding official code as possible on VOC 15-1. The results can be seen in the  response to **Reviewer dzLn ('1.Different joint result of CNN compared with previous methods.')**. These results suggest that our joint result is almost consistent with previous methods, ensuring a fair comparison and to highlight the effectiveness of the proposed methods without modifications to the architecture or minor techniques. We will provide more such reproduced results for other incremental scenarios in final version.
>
> ___
> [1] Saving 100x Storage: Prototype Replay for Reconstructing Training Sample Distribution in Class-Incremental Semantic Segmentation, In NIPS 2023.
>
> [2] Incremental learning techniques for semantic segmentation. In ICCVW, 2019.
>
> [3] Plop: Learning without forgetting for continual semantic segmentation, In CVPR 2021.
>
> [4] Decomposed Knowledge Distillation for Class-Incremental Semantic Segmentation, In NIPS 2022.
>
> [5] Masked-attention mask transformer for universal image segmentation, In CVPR 2022.

---

> > ### Comment · Reviewer_vdnJ · 2025-08-06
> >
> > Thanks for the detailed responses; many of my concerns have been resolved. However, I still have one main concern. To me, the primary difference between CoGaMiD and STAR lies in using a Gaussian mixture model versus a single Gaussian distribution. I remain unconvinced about why the Gaussian mixture model should be used for CISS, beyond the reason of improved quantitative results. In addition, the results for multiple Gaussians tend to increase when K is larger. Please provide more results for K=3, as it would better demonstrate whether Gaussian mixture distribution modeling offers superior performance compared to multiple Gaussian models.

---

> > > ### Author Response · Authors · 2025-08-06
> > >
> > > We thank the reviewer for the feedback.
> > >
> > > ___
> > >
> > > Firstly, we highlighted the differences between our method and STAR in the rebuttal. In addition to using Gaussian Mixture Models, another key distinction is our proposal of a dynamic adjustment strategy to address the mismatch between fixed GMMs and the changing representations due to model updates, which was overlooked in STAR.
> > >
> > >
> > > Secondly, the motivation for adopting Gaussian Mixture Models (GMMs) in CISS has been discussed in our submitted paper. When the incoming data stream for a class is inherently multimodal, using a single prototype may be insufficient to represent the complete distribution of each old class. Several prior works [1,2,3,4] have explored the use of GMMs in class-incremental learning. Inspired by these studies, we are the first to ingeniously incorporate the multimodal nature of GMMs into the CISS framework, enabling more accurate modeling of the distributions of old classes.
> > >
> > > In addition, to provide more results for K>3, following the suggestion of reviewer vPPN, we have conducted the experiments for more values of K, ranging from 4 to 7, as follow:
> > >
> > > **K**|0-15|16-20|all
> > > :-|-|-|-|
> > > 4|79.9|53.2|73.6
> > > 5|79.6|53.1|73.3
> > > 6|79.6|53.5|73.4
> > > 7|79.8|53.3|73.5
> > >
> > > The results reveal a slight decline on performance when K>3. We speculate that using a large number of Gaussian components may lead to partial overfitting, particularly for classes with simple feature distributions in the training set. We believe that K=3 is a relatively reasonable setting, holding performance and computational efficiency simultaneously.
> > >
> > >
> > > [1] Forward compatible few-shot class-incremental learning, In CVPR 2022.
> > >
> > > [2] Learning semi-supervised gaussian mixture models for generalized category discovery, In ICCV 2023.
> > >
> > > [3] Brain-inspired replay for continual learning with artificial neural networks, In Nature Communications 2020.
> > >
> > > [4] Class-incremental mixture of gaussians for deep continual learning, In CVPR 2024.
> > >
> > >
> > >
> > > ____
> > >
> > >  We sincerely thank you for precious time and valuable comments on our work agin.

---

> ### Comment · Area_Chair_mgvj · 2025-08-05
>
> Dear Reviewer vdnJ,
>
> The author-rebuttal phase is now underway, and the authors have provided additional clarifications and performance results in their rebuttal. Could you please take a moment to review their response and engage in the discussion? In particular, we’d appreciate your thoughts on whether their revisions adequately address your initial concerns. Thank you for your time and valuable contributions.
>
> Best, Your AC

---

> ### Comment · Reviewer_vdnJ · 2025-08-06
>
> It would have been helpful to include additional experiments to justify the need for Multiple Gaussians over a Single Gaussian, beyond the observed performance improvements — for example, by examining whether the actual feature distribution is indeed multimodal. In addition, the concept of updating memory when storing features or distributions is not novel, as similar ideas have already been applied in works such as ALIFE [S1]. Furthermore, the proposed Dynamic Adjustment (DA) method itself also appears to lack technical novelty.
>
> Therefore, it would be beneficial to provide further explanations or additional experiments to more convincingly justify the use of GMMs.
>
> [S1] ALIFE: Adaptive Logit Regularizer and Feature Replay for Incremental Semantic Segmentation. NeurIPS 2022.

---

> > ### Author Response · Authors · 2025-08-06
> >
> > We thank the reviewer for the thoughtful feedback.
> >
> > ___
> > **1.Additional experiments to justify the need for Multiple Gaussians over a Single Gaussian, beyond the observed performance improvements.**
> >
> > We believe that the t-SNE results shown in Figure 5 of the paper partially reflect the multimodality of the most classes (e.g. 'person', 'sofa', 'cow', 'train', etc.). These class distributions can be more accurately modeled by Gaussian Mixture Models (GMMs) rather than a single Gaussian. We have also visualized the distributions of each class in the latent space and observed that their shapes are often complex and irregular, making them difficult to approximate with a single Gaussian. We truly want to show these visualizations but there is no way to add new figures during the discussion process. Therefore, we will include a comparative visualization of feature distributions modeled with single and multiple Gaussian components, alongside the real feature distributions, in the later version. This will be accompanied by more thorough analysis to justify the necessity of adopting GMMs.
> >
> > **2.The concept of updating memory when storing features or distributions is not novel, as similar ideas have already been applied in works such as ALIFE [S1]. Furthermore, the proposed Dynamic Adjustment (DA) method itself also appears to lack technical novelty.**
> >
> > Indeed, ALIFE holds a similar view that replayed old features should be updated to remain compatible with the current model.  However, our method is clearly distinct from ALIFE in the way these updates are performed.  ALIFE requires training additional category-specific rotation matrices to update features, which leads to increased computational and memory costs. If the numbers of replayed features is too large, this maner is diffcult to be used in real application. In contrast, we simply update the original parameters of the stored GMMs using old-class features from the current step, thereby avoiding extra memory usage.  Although this process leverages the existing EM algorithm, the underlying idea is both intuitive and effective.
> >
> > ____
> > We sincerely thank you for precious time and valuable comments on our work agin.

---

### Note · Authors · 2025-08-14

Dear Reviewers and ACs,

We sincerely appreciate ACs and all reviewers for their precious time and insightful feedback. The valuable reviews have significantly enhanced the overall quality of the paper.

We are particularly thankful that the reviewers (dzLn, zb2d, vPPN) acknowledge our reasonable identification of issues in single-prototype-base CISS methods and our effective approach to addressing these issues. Additionally, we are grateful for believing that our approach can assist a segmentation model in mitigating forgetting (vdnJ) and our paper is well written (dzLn). Finally, we are thankful for noting our paper show promising and strong performance (dzLn, zb2d, vPPN).

We have responded to all four reviewers individually during the rebuttal and discussion phases. The progress we have made in addressing most of concerns can be summarized as follows:
- Additional results for different K values: Following the constructive suggestions of reviewers (vdnJ, vPPN), we added the more results for different values of K to not only further demonstrate effectiveness of using multiple Gaussians but also provide deeper analysis and discussion on the selection of K.

- Joint training results: In response to thoughtful comments from reviewers (vdnJ, dzLn), we reproduced as many results as possible for both incremental learning and joint training using the corresponding official code to show the fairness of our comparison.

- Additional qualitative results: Based on the insightful feedback from reviewers (vdnJ, dzLn), we conducted visualization experiments to illustrate the multimodal of feature distribution and the anisotropy of features during the incremental steps.

- Additional ablation and comparative experiments: We carried out futher ablation studies to clarify the previously unclear notation of $L_{kd}$, and to further validate effectiveness of our method (vdnJ, zb2d). We included additional comparisons regarding computational cost and Mask2Former-based architectures to support the superiority of our approach (dzLn).

- Further clarification and modification of typos: Following the comprehensive recommendations of reviewers (vdnJ, zb2d, vPPN), we clarified several aspects of our approach (e.g. main distinctions compared to other methods, usage of GMM features, settings of the baseline) and corrected all identified typos.

We sincerely thank you again for the valuable feedback, which we believe have significantly improved the paper.

Best Regards,

Authors

---

### Decision · Program_Chairs · 2025-09-17

**Decision:**

Accept (poster)

**Comment:**

The paper proposes a class-incremental semantic segmentation method that models feature distributions using Gaussian Mixture Models (GMMs) to mitigate catastrophic forgetting, enhanced by a dynamic adjustment strategy for refining old-class features and a Gaussian-based loss to maintain discriminability between old and new classes.

The reviewers acknowledged several strengths of the proposed method: 1) a novel approach using Gaussian Mixture Models (GMMs) to generate features for previously learned classes in class-incremental semantic segmentation; 2) effective design of its dynamic adjustment strategy  for coping distribution shifts during incremental updates and its Gaussian-based representation constraint loss for preserving class separability between old and new classes; 3) strong empirical performances on Pascal VOC and ADE20K, particularly in long-term incremental settings. However, they also raised significant concerns initially regarding: 1) limited technical novelty w.r.t. prior work (STAR); 2) Insufficient experimental analysis on the effectiveness of using multiple Gaussians (K=3) and lack of experiments comparing computational and memory costs with previous methods; 3) unclear mathematical formulations and insufficient explanation of how the segmentation network integrates the GMM features during classification; and 4) discrepancies in baseline results compared to previous work, raising questions about fair comparison.

In their rebuttal discussion, the authors provided a detailed response including 1) further clarifications on novelty w.r.t. related work and technical design, 2) additional ablation (e.g., K values), computational cost results and experimental comparisons, 3) reproduced joint training results to show the fairness of our comparison.  The rebuttal addressed most of the reviewers' concerns, and all four reviewers responded positively with three of them raising their scores. Specifically, Reviewers dzLn and zb2d recommended acceptance, while the other two reviewers gave a positive assessment, conditioned on including the additional qualitative analysis of feature distributions.

The AC agrees with the reviewers' overall assessment: the paper presents a well-motivated and empirically effective approach to class-incremental semantic segmentation, demonstrating strong performance. While questions remain regarding the novelty/necessity of its GMM-based design, the paper’s technical contributions and experimental results justify acceptance. The authors are expected to address all reviewer concerns—including those related to novelty, GMM motivation, and technical clarity—in the final revision, incorporating insights from their rebuttal.